# Pretreatment-free SERS sensing of microplastics using a self-attention-based neural network on hierarchically porous Ag foams

Olga Guselnikova [1,2] ✉, Andrii Trelin[3], Yunqing Kang[1,4], Pavel Postnikov [2,3], Makoto Kobashi [4], Asuka Suzuki[4], Lok Kumar Shrestha [1,5], Joel Henzie [1] ✉ & Yusuke Yamauchi[4,6] ✉

Low-cost detection systems are needed for the identification of microplastics (MPs) in environmental samples. However, their rapid identification is hindered by the need for complex isolation and pre-treatment methods. This study describes a comprehensive sensing platform to identify MPs in environmental samples without requiring independent separation or pre-treatment protocols. It leverages the physicochemical properties of macroporous-mesoporous silver (Ag) substrates templated with self-assembled polymeric micelles to concurrently separate and analyze multiple MP targets using surface-enhanced Raman spectroscopy (SERS). The hydrophobic layer on Ag aids in stabilizing the nanostructures in the environment and mitigates biofouling. To monitor complex samples with multiple MPs and to demultiplex numerous overlapping patterns, we develop a neural network (NN) algorithm called SpecATNet that employs a self-attention mechanism to resolve the complex dependencies and patterns in SERS data to identify six common types of MPs: polystyrene, polyethylene, polymethylmethacrylate, polytetrafluoroethylene, nylon, and polyethylene terephthalate. SpecATNet uses multi-label classification to analyze multi-component mixtures even in the presence of various interference agents. The combination of macroporous-mesoporous Ag substrates and self-attention-based NN technology holds potential to enable field monitoring of MPs by generating rich datasets that machines can interpret and analyze.

The plastics industry generates ≈400 million tons of polymer materials annually[1]. A large portion is eventually deposited in oceans and waterways, where it becomes divided into tiny fragments <5 mm, commonly called microplastics (MPs)[2]. Inexpensive detection systems capable of identifying MPs in marine and freshwater environments are needed to locate sources of MPs and anticipate where MPs could have consequential effects on public health[1]. However, rapidly identifying MPs in aqueous environmental samples is challenging because: (i) MPs

---

[1]National Institute for Materials Science (NIMS), Tsukuba, Ibaraki, Japan. [2]Research School of Chemistry and Applied Biomedical Sciences, Tomsk Polytechnic University, Tomsk, Russian Federation. [3]Department of Solid-State Engineering, University of Chemistry and Technology, Prague, Czech Republic. [4]Department of Materials Process Engineering, Graduate School of Engineering, Nagoya University, Furo-cho, Chikusa-ku, Nagoya, Japan. [5]Department of Materials Science, Institute of Pure and Applied Sciences, University of Tsukuba, Tsukuba, Ibaraki, Japan. [6]Australian Institute for Bioengineering and Nanotechnology (AIBN), The University of Queensland, Brisbane, QLD, Australia. ✉e-mail: guselnikovaoa@tpu.ru; henzie.joeladam@nims.go.jp; y.yamauchi@uq.edu.au

have similar chemical structures, (ii) MPs exist at low concentrations in the environmental matrix versus other kinds of organic and inorganic materials, and (iii) long-term exposure to the environment can modify the surfaces of MPs to obfuscate their chemical structure[2,3]. As a result, conventional approaches require the isolation of MPs from the environmental matrix, separating them into manageable components through various preconcentration (i.e., sedimentation, sieving, etc.) and chemical pretreatment (i.e., chemical digestion, etc) methods[3]. These preconcentration and pretreatment procedures, henceforth abbreviated as "PCPT", are a significant bottleneck in MP sensing throughput because complex mixtures typically require 12–24 h[4,5] before analysis can begin. Eliminating PCPT steps from a sensing workflow would accelerate the detection of MPs[6]. Additionally, creating an inexpensive MP sensing workflow that leverages open-source automated MP classification tools and instruments would open up MP sensing even to resource-limited labs.

To create a sensor that omits PCPT methods, the sensor should demonstrate the ability to sense individual particles while having some built-in affinity for MPs. MPs are organic macromolecules that tend to accumulate on hydrophobic surfaces and are sufficiently large enough to be strongly governed by capillary forces. To sample the uppermost layer of water bodies for organisms and particles, researchers use specialized equipment like Neuston Nets. These nets are designed to passively collect samples by combining physical sieving with capillary forces, aiding in the study of aquatic environments[7]. But Neuston nets have no inherent sensing ability, so the MPs collected by nets and other collection methods are subjected to laborious PCPT methods and then passed to a method with an analytical readout like visual analysis, thermal/mass spectroscopy, or optical spectroscopy (Suppl. Table 1). Visual analysis methods can use stereomicroscopes to sort and identify MPs, but visual methods are susceptible to false-positive/negative results as particle sizes decrease[8]. In contrast to visual, mass spectroscopy tools like pyrolysis-gas chromatography mass chromatography (pyr-GCMS) can provide chemical information on the polymers present but require expensive equipment that is not portable and needs highly qualified operators. Optical spectroscopy methods like Fourier-transform infrared (FTIR) or Raman spectroscopy are useful because they can identify the chemical structure of MPs and are relatively inexpensive and portable compared to mass/thermal spectroscopy tools (See Suppl. Table 1 for a comparison of different methods). Raman spectroscopy has better spatial resolution than FTIR[2], lower water interference, and narrower spectral bands. However, the analytical identification of MPs with Raman spectroscopy is still limited because their Raman peaks overlap, and the polarizability of most polymers is relatively weak and generates autofluorescence[9]. Surface-enhanced Raman spectroscopy (SERS) is a complementary technique that can achieve single-molecule sensitivity by coupling light into the collective oscillations of free electrons called surface plasmons (SP) on the surface of nanostructured noble metals[10]. This form of optical confinement enabled by the metal generates intense local electric field intensities ($|E|^2$) corresponding to strong SERS signals from adjacent molecules. However, $|E|^2$ decays exponentially orthogonal to the metal/dielectric interface[11], thus the excitation of micron-scale analytes like MPs with SPs is challenging using conventional nanostructured metal surfaces[12] because only a small section of the particle is excited by the localized electric field (Fig. 1a). In addition, these surfaces generate minimal capillary forces, limiting their ability to trap large MPs from flowing aqueous solutions. Researchers have made micro-fabricated quasi-3D plasmonic optical grating structures with features in the same order as the SERS excitation wavelength[13]. However, these methods are unsuitable for MPs > 1 μm, which is more typical in water-bourne particles and would still require PCPT methods for complex samples.

We hypothesized that SERS could be a viable route to a PCPT-free sensing method for MPs if the length scale of the metal substrate has macroscale features and physicochemical affinity to recruit MPs from aqueous solutions while still supporting local electric fields to generate strong SERS signals (Fig. 1b). In addition, the surface of the metal must be sufficiently hydrophobic to recruit MPs from solution and help amplify capillary forces to trap the MPs in a tortuous macroporous network. Environmental samples contain numerous charged organic and inorganic agents in addition to MPs so this structure should be capable of separating or demultiplexing the MPs from a flowing aqueous solution (Fig. 1c). The 3D metal structure would also create a more volumetric-like plasmon mode where the MPs can be excited on all sides by the SP and scattered light for SERS detection of adsorbed species. Still, complex samples containing numerous kinds of MPs and other interference agents will generate complex Raman spectra with overlapping peaks that are difficult to interpret (Fig. 1d), thus porous structure alone is unlikely to enable PCPT-free SERS. Recently, researchers have applied neural networks (NNs) to various analytical detection techniques to make consequential decisions about the makeup of unknown samples[14,15]. For example, NNs have enhanced SERS detection techniques to analyze complex mixtures, including wine and photodamaged DNA[16,17]. But conventional convolutional NNs (CNNs) begin to deliver lower accuracy judgments with complex and nuanced data that contains long-range dependencies. Simple CNNs can be upgraded using a self-attention mechanism borrowed from natural language processing (NLP) models called Transformers[18]. Self-attention can enable a CNN to simultaneously weigh the relationship between the input data points relative to any part in the sequence and interpret complex dependencies[18,19]. Adding self-attention into a CNN architecture for SERS (Fig. 1d) should enable the model to identify spectra containing MPs and allocate more significance or weight to this data when determining the presence of different MPs. This paper describes a sensing workflow that combines porous 3D SERS substrates with self-attention-based CNN to physically and computationally demultiplex SERS spectra. We tested the workflow on complex mixtures of MPs and various environmental interference agents to accurately determine the presence of 6-types of MPs (i.e., polystyrene, polyethylene, polymethylmethacrylate, polytetrafluoroethylene, nylon, and polyethyele terephtalate).

## Results and discussion

### Macro-mesoporous Ag foams coated with a hydrophobic layer

Concave surfaces can exert strong capillary forces on microscale objects suspended in water[20,21]. We hypothesized that convoluted porous metal foams with hydrophobic surfaces could play a dual role of trapping MPs from solution to facilitate the omission of PCPT techniques while generating SPs to enable SERS. Light can excite SPs on Ag surfaces and propagate tens to hundreds of microns[11], enabling the excitation of MPs trapped deep inside a tortuous network of macroporous and mesoporous metal nanostructures[22]. We initially considered open-cell macroporous metal foams because they are widely available in various metals (e.g., Ag, Al, Ni, Ti, etc.) and can be modified by electrodeposition methods to generate various 3D mesoporous metals[23]. We previously developed an inexpensive and scalable method to deposit mesoporous metals (e.g., Au, Ag, Cu) on conductive 2D electrodes by co-depositing block copolymer micelles (BCM) with metal using a simple electrochemical setup[23]. The size of the mesopores can be tuned from 5 to 40 nm in diameter depending on the diameter of the BCM, and the resulting subwavelength pores interact strongly with light to enable SERS sensing[24,25]. In an adapted procedure, the commercial 3D silver foam (AgF) was used as the electrode, enabling $Ag^+$ and polystyrene$_{18000}$-block-polyethylene oxide$_{7500}$ (PS$_{18000}$-$b$-PEO$_{7500}$) BCMs to be co-deposited on the 3D surface (Fig. 2a and Suppl. Fig. 1). According to X-ray micro-tomography (μ-CT), unmodified AgF have convoluted macroscale features with an average pore diameter of 262 μm with a void space of 84.1% versus the total volume (Fig. 2b and Suppl. Fig. 2). SEM images

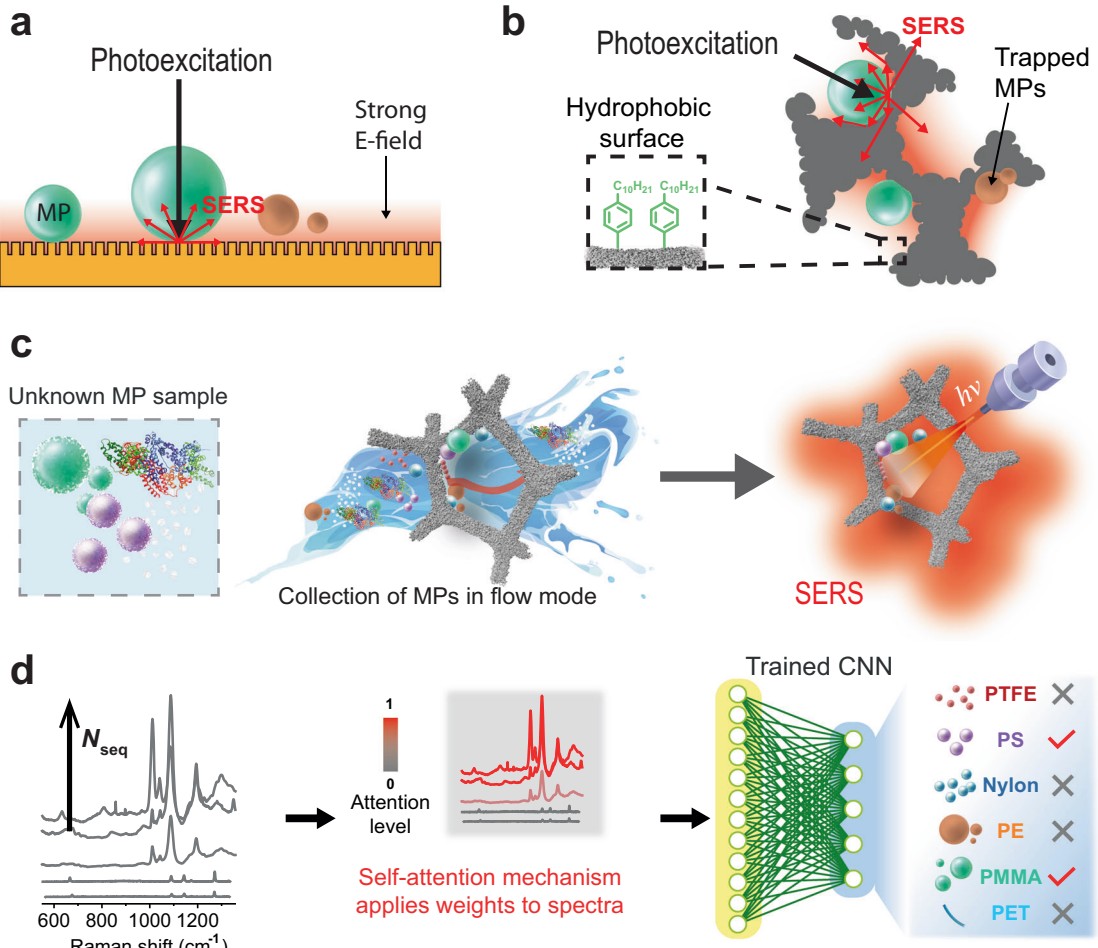

**Fig. 1 | Strategy for detection of MPs using porous plasmonic substrates and a self-attention-based NN. a** MPs dispersed on plasmonic metal gratings experience strong electric fields, but mainly within the small contact points between the particle and grating. Capillary forces are also relatively weak on subwavelength gratings, limiting its ability to collect large (>1 μm) MPs from flowing liquids. **b** Macroporous metals have large convex/concave surfaces and an open framework that enables both light and MPs to penetrate deep into the metal network. Texturing the metal surfaces with mesopores enables light to excite intense local electric field intensities that correspond to strong SERS enhancements in adjacent molecules. Coating the surface with hydrophobic groups facilitates the trapping of MPs versus small water-soluble molecules. **c** MPs in environmental samples are relatively dilute compared to other forms of organic and inorganic matter thus, preconcentration and pretreatment methods are typically necessary before analytical measurement. Our macroporous metals are designed to recruit MPs from aqueous solution and prevent biofouling. The metal surface also promotes strong electric field intensities for SERS. (water splash by @pch.vector reproduced with permission from www.freepik.com, "water splashes flat icon set"; the protein illustration is based on refs. 76,77) **d** batches of SERS spectra are collected on the metal sample to obtain the chemical information of the sample. Then a NN called SpecATNet uses a self-attention mechanism to assign weights to the data. The weights enable the NN to essentially pay more attention to MP-related spectra and ignore less pertinent data, allowing the model to more accurately determine the presence of various MPs even in complex mixtures.

show the surface of the AgF at various magnifications and how it is coated with a mesoporous Ag film (AgM) with an average pore size of 28 ± 4 nm in diameter (Fig. 2c–e). This structure is called AgF@AgM to indicate that the mesoporous Ag metal encapsulates the AgF (Fig. 2). The electrochemical active surface area (ECSA) of the Ag foam increased by 25% after the formation of the mesoporous Ag layer (Suppl. Fig. 3). This deposition method is an example of noble metal mesopores deposited directly on large macroscale 3D metal structures.

Hydrophobization of AgF@AgM could increase capillary forces and promote hydrophobic interactions to drive the selectivity toward MPs. Moreover, the hydrophobic coating could improve the long-term stability of the Ag surface by limiting oxidation[26], while also mitigating biofouling by small biomolecules[27,28]. Diazonium-based surface chemistry was used to coat the AgF@AgM substrate with $a \approx 2.4$ nm thick monolayer of hydrophobic 4-decylphenyl (C10) groups to form the AgF@AgM@C10 sample (Suppl. Note 1). Diazonium salts spontaneously interact with noble metal surfaces *via* a de-diazonation reaction, forming reactive aryl radicals that covalently attach to the silver surface *via* Ag-C bonds[29]. We also used this diazonium reaction to form hydrophilic AgF@AgM@COOH by attaching 4-carboxyphenyl groups, as demonstrated with x-ray photoelectron spectroscopy (XPS) and Raman spectroscopy (Suppl. Figs. 4 and 5, Suppl. Table 2, and Suppl. Note 1). The addition of the C10 monolayer made the Ag surface hydrophobic, generating a water contact angle (WCA) of 162 ± 2.8° and a surface free energy (SFE) of 30.6 mJ m$^{-2}$ (Suppl. Fig. 6 and Suppl. Note 2). But water can still penetrate deep into hydrophobic micro-structured or pillared substrates depending on the local topographical structure of the surface[30]. We flowed solutions containing PS particles over the substrates using a peristaltic pump to test how water and MPs interact with the AgF@AgM@C10 substrate. SEM images of a cross-section of the AgF@AgM@C10 structure show that the PS particles penetrate deep into the interior of the foam and are randomly distributed throughout the hydrophobic metal network (Suppl. Fig. 7). We found that the AgF@AgM@C10 substrate could capture ≈20× more PS versus the original AgF foam sample (Suppl. Fig. 8).

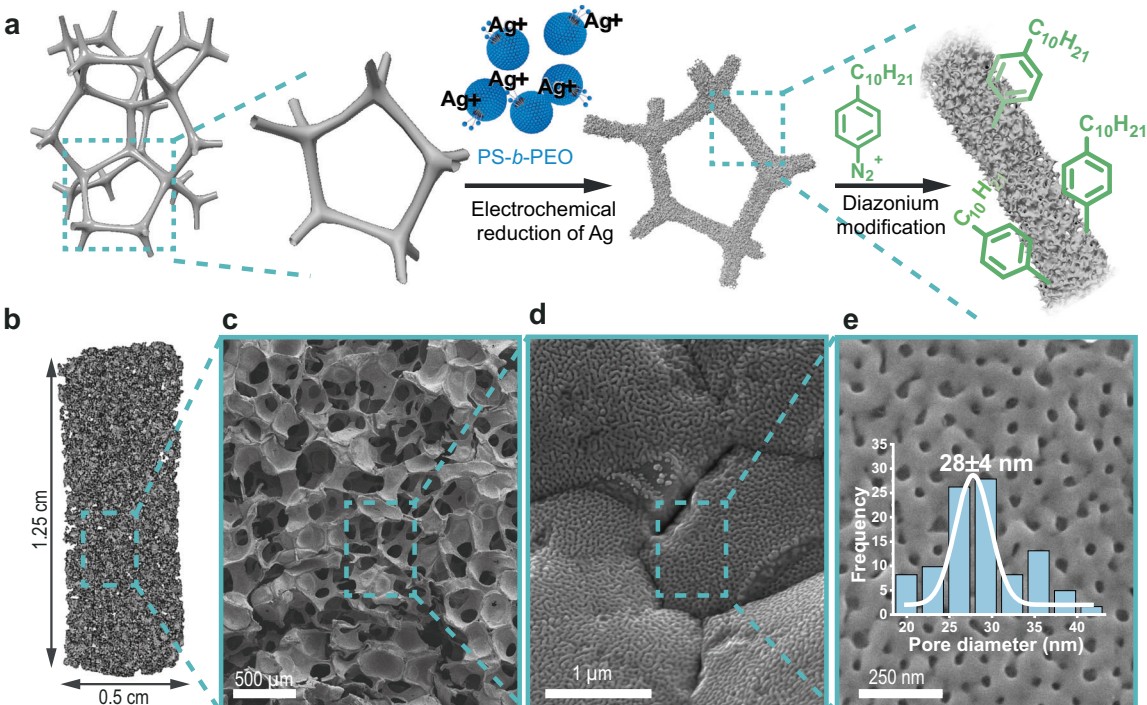

**Fig. 2 | Preparation and characterization of AgF@AgM@C10. a** An illustration describing the electrodeposition method used to coat porous Ag foam with mesoporous Ag to make AgF@AgM. After the co-deposition of Ag and BCMs, the mesoporous Ag surface is passivated with 4-decylbenzene diazonium tosylate to make the films hydrophobic. The passivated substrates are called AgF@AgM@C10. **b** The AgF was imaged with μ-CT to generate 3D models of the macroporous foam network. The average diameter of the pores is 262 μm, and the void space occupied 84.1% of the total volume of the material. **c–e** SEM images of the AgF@AgM structure at increasing magnification. The pore size distribution of the mesoporous Ag was estimated using SEM and had an average pore diameter of 28 ± 4 nm.

Substitution of the hydrophobic @C10 coatings with hydrophilic @COOH carboxy coatings (Suppl. Fig. 9) decreased the PS adsorption capacity by ≈3×. Under identical conditions, the commercial SERS substrate Klarite 313 captured 43× fewer PS particles than AgF@AgM@C10. Modifying the AgF substrate with mesopores and hydrophobic groups amplifies capillary and hydrophobic forces for preferential interactions with MP in water rather than absorbing small charged biomolecules[31]. Simulations can help explain the impact of mesoporosity of permeability and fluid velocity to help explain the ability of AgF@AgM to capture MPs. The μ-CT models of AgF were artificially textured to mimic the mesoporous surface of AgF@AgM since μ-CT had insufficient resolution to resolve mesoporosity. The introduction of texture to the surface of the model decreases permeability and slows fluid velocity (Suppl. Fig. 10). Smaller and more tortuous pore pathways are expected to decrease permeability, allowing capillary forces to become more significant in mass transport, and leading to greater retention of fluids and MPs. Slower fluid velocity also enhances capillary forces because capillary forces have more opportunity to distribute the fluid in the porous medium, in addition to enabling other forces like gravity to impact MP retention.

**Optical properties of AgF@AgM foams and SERS sensing of MPs**
Introducing short-range order in disordered structures can significantly enhance light absorption[32]. The resulting mesoporous structures create short-range order on the surface of the disordered AgF, which should enhance light coupling to SP modes and increase the probability that an adjacent molecule or MP couples with the SP to generate a Raman scattered photon for SERS sensing. We measured the AgF and AgF@AgM samples in UV-VIS (Suppl. Fig. 11), and both samples have an absorbance peak at ≈315 nm that matches the interband transitions of Ag. The addition of the mesoporous Ag generates a broad resonance with a peak at 360 nm that corresponds to the plasmon resonance. A SEM image of a flat section on the AgF@AgM surface

was taken and input into a 3D electromagnetic (EM) simulator (Fig. 3a). Ag-based SERS sensors frequently use $\lambda = 532$ nm excitation since it overlaps with the plasmon resonance. At 532 nm, the electric field intensity ($|E|^2$) near the surface of the mesoporous Ag film has various EM hotspots with locations and intensities that depend on the light polarization (Fig. 3b and Suppl. Fig. 12). The simulated reflectance spectrum has a dip at ≈310 nm corresponding to the interband transitions (ITB) of Ag, in addition to a slight dip at 370 nm, and two larger dips at 410 and 515 nm that likely correspond to SP modes (Fig. 3c). The average electric field intensity at each wavelength was plotted in Fig. 3c and had no peak corresponding to the ITB, but the peaks at 370, 410 and 515 nm match the reflectance spectra indicating these are LSPRs. We also examined how the macroporous Ag foam interacted with light by inputting a 478 μm × 560 μm × 450 μm section of the μ-CT scan into the EM simulator. It was excited with a $\lambda = 532$ nm Gaussian beam because that is most similar to a confocal SERS setup (Fig. 3d). The green circle indicates the approximate location of the Gaussian beam. The polarization plot shows how the metal surface becomes polarized upon excitation, with charge accumulating along the concave and convex features of the foam. The simulations indicate that the AgF@AgM structures strongly couple with light and support SP resonances $\lambda < 600$ nm. Using metals like gold or copper should facilitate SP resonances at longer wavelengths.

Flat mesoporous Ag films coated with methylene blue (MB, $\lambda_{abs} = 665$ nm) produced SERS enhancement factors (EF) $> 10^5$ when excited with $\lambda = 532$ nm (Fig. 3e)[33]. In contrast, AgF generated negligible SERS signal under the same conditions. Coating the AgF with mesopores (i.e., AgF@AgM) generated an EF of $2.5 \times 10^5$ EF, ostensibly because mesopores generate strong EM hotspots and the convoluted surface enhances coupling due to the sensitivity of the LSPR to the wavevector of light[34]. We mapped the SERS intensity of MB at 1637 cm$^{-1}$ and observed the strongest signals at open voids created in the foam (Suppl. Fig. 13). The AgF@AgM@C10 sample generated the strongest

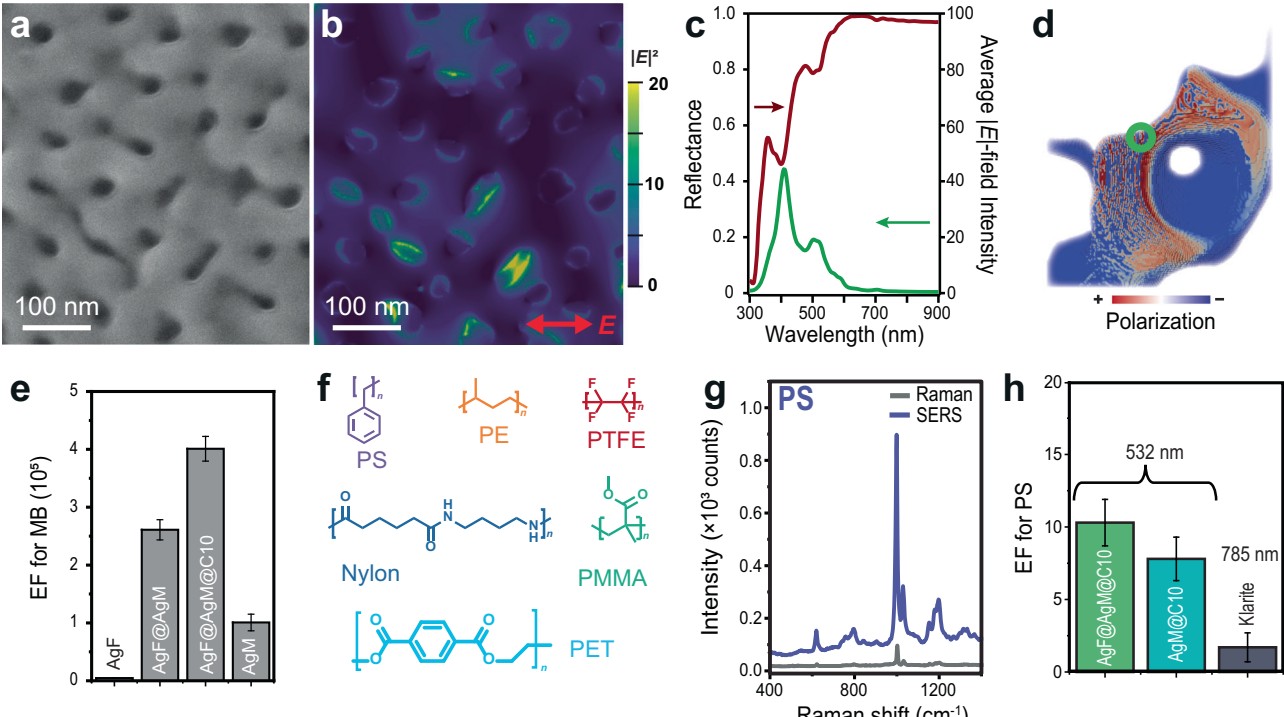

**Fig. 3 | Optical properties and plasmonic performance of AgF@AgM@C10.**
**a** SEM image of the AgF@AgM@C10. **b** The simulated $|E|^2$ distribution of the SEM image in **a** at $\lambda = 532$ nm. **c** Simulated reflectance spectra and corresponding electric field spectrum as a function of wavelength. **d** Simulated charge polarization map of plasmon on one pore of AgF@AgM using Gaussian beam (indicated by the green circle). **e** Comparison of the EF for MB obtained on AgF, AgF@AgM, AgF@AgM@C10, AgM. **f** An illustration showing six types of MPs used in this study: PS, PE, Nylon, PMMA, PTFE, and PET sized by 300–90 μm. **g** SERS spectra of PS on glass and AgF@AgM@C10 after manual focusing. **h** A comparison of EF for PS was obtained on AgF@AgM@C10 and AgM@C10 (at 532 nm) and Klarite (at 785 nm). For **e** and **h**, the data represents the mean value with residual standard deviation calculated from 3 measurements on 3 different samples ($N = 3$, $n = 3$).

SERS EF ($4.0 \times 10^5$) observed versus the other samples. Although the @C10 decylphenyl layer increases the distance between MB and the Ag metal, which should reduce SERS EF due to the lower EM field further from the surface, the hydrophobic surface helps recruit hydrophobic molecules like MB and also limits Ag oxidation[35,36]. The SERS signal of MB is 12× more stable on AgF@AgM@C10 than on AgF@AgM after storage in air for 1 month as seen on Ag 3*d* XPS region (Suppl. Fig. 14) or exposure to water for 3 days (Suppl. Fig. 15).

The AgF@AgM@C10 sample supports a strong SERS response and is compatible with hydrophobic molecules. To test the AgF@AgM@C10 as an MP detection system we selected the following 6 polymers based on their role in both industrial-derived and consumer-derived pollution sources, their ability to disperse in fresh and marine environments, in addition to their diversity of chemical structure, size, and shape (see Suppl. Fig. 16 and Suppl. Note 3): 30 μm polystyrene (PS), 10-90 μm polyethylene (PE), 300 nm polytetrafluoroethylene (PTFE), 27-45 μm polymethylmethacrylate (PMMA), 3 μm Nylon, and 20 μm thick polyethylene terephthalate (PET) fiber of different lengths[37] (Fig. 3f and Suppl. Fig. 17). The MPs were quasispherical in shape and covered a wide size range from 300 nm to 90 μm (see Suppl. Fig. 17 for SEM analysis of sizes and morphology). Figure 3g shows manually collected Raman and SERS spectra for the 30 μm PS particles deposited on glass versus AgF@AgM@C10. The peak at 1005 cm$^{-1}$ appears in both the Raman (glass) and SERS (AgF@AgM@C10) spectra. We fabricated a mesoporous Ag film on SiO$_2$ and coated it with decylphenyl groups (AgM@C10, Suppl. Fig. 18). When using glass as the baseline, flat AgM@C10 generated 5× more SERS signal than glass. AgF@AgM@C10 shows the highest EF of 10× because the volume of the EM-field in pores is large and generates bigger Raman absorbance than the flat AgM@C10 surface (Fig. 3h). These findings confirm the critical role of macropores in enhancing

Raman scattering of MPs. Finally, commercial Klarite substrates were used to examine the SERS signals from PS MPs. These substrates use Au so they were excited using 785-nm excitation. Klarite generated a lower EF at 785 nm than AgF@AgM@C10 at 532 nm. The EF values obtained for Klarite are lower compared to the results reported by Zhang et al. [12] for smaller PS beads (0.36 to 5 μm), likely because the 30 μm PS particles cannot fit into the inverted pyramidal gratings of the Klarite sensor. Similar measurements were conducted for PE, PTFE, PMMA, and Nylon (Suppl. Figs. 19–21 and Suppl. Table 3). Each polymer generates a characteristic array of peaks on the AgF@AgM@C10 substrate. Nonetheless, the same trend in EF was observed, with AgF@AgM@C10 showing the highest EF followed by AgM@C10, Klarite, and glass.

Researchers typically collect MPs in the environment by trawling through polluted water using Neuston nets or equivalent. These nets were initially designed for collecting plankton and can sieve large volumes of flowing water[38]. The 3D porous structure of the AgF@AgM@C10 is intended to be like a Neuston net capable of trapping MPs from flowing solutions using a combination of capillary and hydrophobic forces. MPs adsorb and accumulate on the surfaces of AgF@AgM@C10 due to hydrophobic and capillary forces. We initially flowed suspensions of PS ($1.06 \times 10^4$ particles per liter or MPs L$^{-1}$) and PE ($1.9 \times 10^4$ MPs L$^{-1}$) through the AgF@AgM@C10 substrate (Suppl. Fig. 22) for 20 min. Later the substrates were dried, and SERS was collected using a confocal Raman microscope. The confocal Raman microscope generates large 2D survey maps of an AgF@AgM@C10 substrate where MPs are distributed randomly through the interior and exterior of the foam (Suppl. Fig. 23). The SERS peaks associated with PS and PE vary over time due to: (i) variations in the locations of the MPs on the Ag surface, (ii) changes in the polymer spectra due to local gyration of the molecular substituents within the

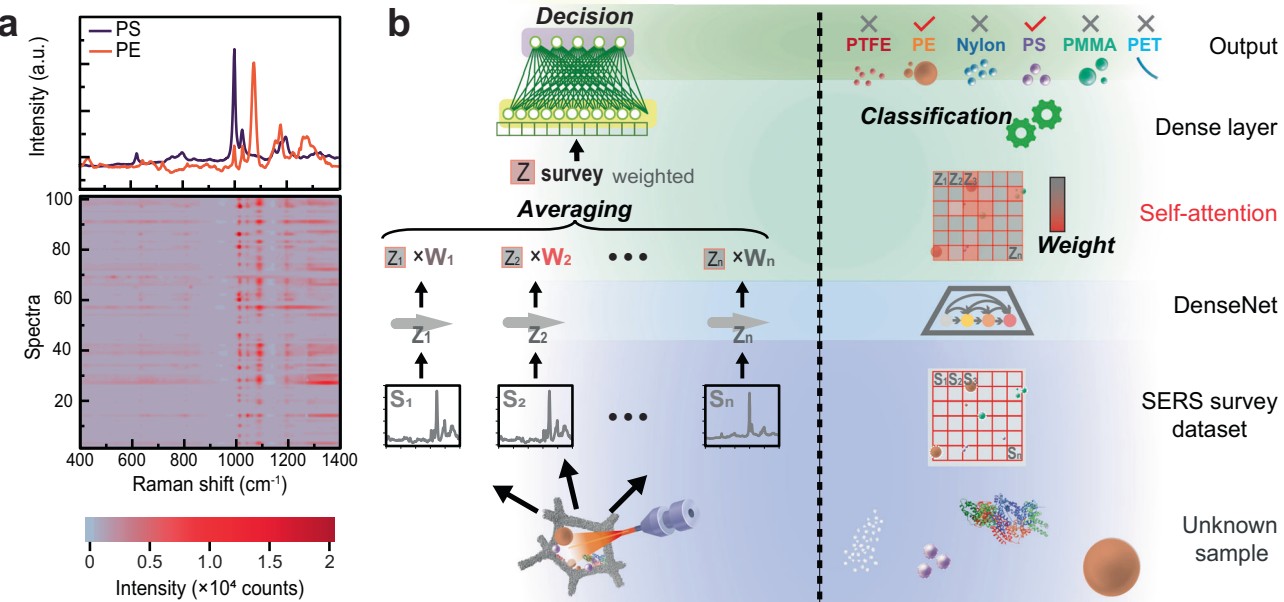

**Fig. 4 | SpecATNet for the analysis of MPs mixtures. a** Waterfall plot of sequential SERS spectra for a PE/PS mixture (the violet trace represents the typical spectrum of PS, while the orange trace represents the typical spectrum of PE). **b** The architecture of developed convolutional NN SpecATNet, with a self-attention function used to solve a multi-label classification task for 6 polymers (from bottom to top). SERS spectra from unknown samples are collected in the survey dataset.

These surveys are pre-processed, submitted to a DenseNet, and converted into encoded pixels ($Z_1, Z_2... Z_n$) for pattern extraction. Self-attention is used to weight ($W_1, W_2... W_n$) the encoded pixels to generate $Z_{survey}$ (averaged summary). A Dense Layer makes the multi-label classification decisions (the protein illustration is based on refs. [76],[77]).

EM hotspots and (iii) overlapping peaks from different MP types (Fig. 4a). These dynamics make SERS data challenging to interpret because it requires the ability to accurately process hundreds or even thousands of spectra to judge the composition of multi-component MP mixtures[39].

### Self-attention-based CNN to identify MPs in complex mixtures

The physicochemical interactions between the MPs and AgF@AgM@C10 substrate help separate or demultiplex the SERS signal from the solvent and interference agents, but these interactions are not specific enough to enable SERS to identify complex mixtures of MPs and other interfering agents with a high level of confidence. Deep Learning (DL) models identify patterns in unstructured data and make statistical decisions based on these patterns[16,17]. Convolutional NNs (CNNs) are a class of DL models that can adaptively learn spatial hierarchies of features from input data by learning from examples. Previously, CNNs were used to classify the presence/absence of polymers based on vibrational data[14,40–42]. However, this work mainly uses binary or multi-class classification, which is sufficient when the prediction can be one class of materials but has limited accuracy with complex multi-component samples collected in the environment. Multi-label classification methods are more suitable for classifying complex mixtures with multiple polymers.

The AgF@AgM@C10 SERS substrates act like hydrophobic sieves to collect MPs from flowing solutions (Suppl. Fig. 22). Moreover, the complex surface of the metal helps capture the dynamic nature of MPs via SERS by exposing the MP ensemble to a wide range of excitation conditions. This effect helps ensure that the training dataset given to SpecATNet minimizes time and orientation-dependent inhomogeneity of the SERS signals, giving a more realistic representation of the MP ensemble (Fig. 4a and Suppl. Fig. 23). In a SERS survey map, each point ($S_1, S_2... S_n$) is one spectrum and thus contains information on some unknown fraction of the total chemical composition of the sample (Fig. 4b). In standard CNN classification tasks, it is assumed that a class can be determined from

a single spectrum[43]. Alternatively, the Bayesian decision rule[44] can combine information from multiple spectra.

We employed an extension to the CNN architecture we call SpecATNet (Fig. 4b), which can consume multiple SERS spectra and make predictions using the combined information. Each spectrum in the SERS survey is pre-processed with a dense CNN (DenseNet) to transform it into hidden representations or simplified encoded pixels ($Z_1$, $Z_2... Z_n$) that highlight the amplified chemical features and together provide a condensed summary of the sample ($Z_{survey}$). DenseNet was chosen as a typical DL NN due to its slightly higher performance versus other DL methods (Suppl. Fig. 24). Still, any NN that helps omit PCPT methods in SERS identification of MPs is the primary goal of this work, so experimentation with NNs in the future could be fruitful. Some locations on the sample contain no MPs or multiple MPs, thus the model must be trained to apply a weight ($W_1, W_2... W_n$) to each corresponding encoded pixel to reflect the importance of the underlying chemical information. In other words, applying weights to the data trains SpecATNet to pay more attention to some data and effectively ignore less relevant data—this is a guiding principle of the self-attention mechanism used more prominently for NLP[18]. We modified the self-attention mechanism to assign weights ($W_1, W_2... W_n$) to every measured survey point ($Z_1, Z_2... Z_n$) for proper averaging of spectra and generating a reliable $Z_{survey}$ to make decisions about the presence of each MP. Assigning weight via self-attention within SpecATNet training is advantageous compared to previously described manual averaging and post-CNN statistical approaches[43,44] due to adaptivity to different samples. The output of SpecATNet is the probabilities [0 to 1] of the presence of each MP in the unknown sample visualized later as a confusion matrix.

### SpecATNet for MP samples with increasing multiplexity

To evaluate the predictive performance of SpecATNet, we tested it using samples with increasing multiplexity. In this context, multiplexity refers to the increasing number and types of MPs present in each sample (Suppl. Fig. 25). The lowest multiplexity sample contains a

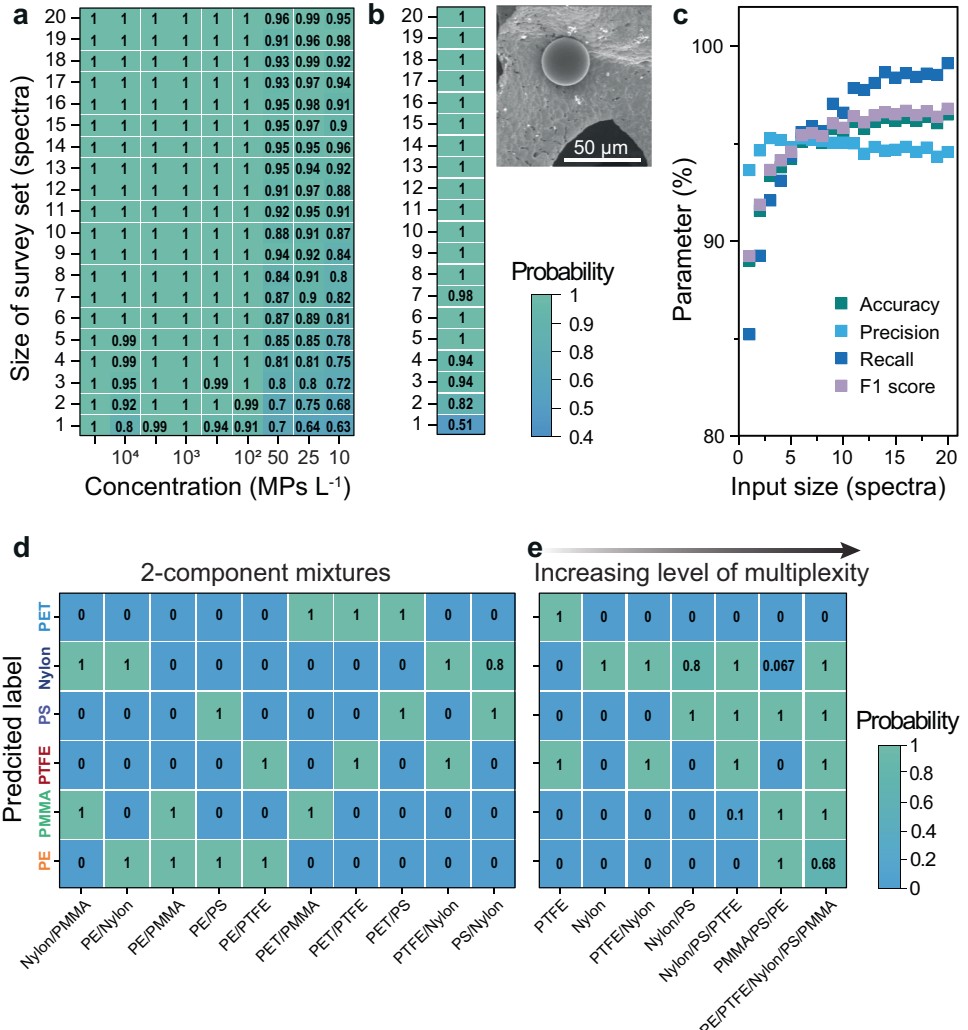

**Fig. 5 | AgF@AgM@C10 and SpecATNet for analysis of MPs of different complexity. a** SpecATNet predictions as a function of concentration and number of spectra for a PS suspension ($10^1$–$10^4$ MPs L$^{-1}$), and **b** for the detection of a single PS particle. **c** A chart showing how the accuracy, precision, recall and F1 change as the number of input spectra is increased for a single PS particle. **d** SpecATNet predictions presented in a confusion matrix showing the classification of two-component mixtures of MPs and **e** multi-component mixtures with varying levels of multiplexity. The row and column indices correspond to the ground truth and predicted label, respectively.

single MP. We flowed 6 different solutions containing one of the MPs in water over the AgF@AgM@C10 substrates and collected the SERS survey maps (see concentrations in Suppl. Table 4). The SERS surveys were input into SpecATNet, which assigned a probability with 100% accuracy that each MP was present (Suppl. Figs. 26 and 27a). For comparison, ML-based methods using vibrational spectroscopy, such as support vector machines, have ≈94% accuracy[42,45], or 96% accuracy for CNNs analyzing single MP samples (Suppl. Table 5)[46]. Lower accuracy is likely explained by dataset quality; SERS provides high sensitivity and intensity of spectra compared to Raman and FT-IR[42,45,46]. The structure of the AgF@AgM substrates combined with SpecATNet enables high accuracy and the ability to collect and identify MPs without PCPT methods.

Next, we examined the AgF@AgM@C10 substrates at low concentrations of PS MPs to determine under what conditions SpecATNet can detect MPs in concentrations found in real environmental samples (i.e., $10^4$ MPs L$^{-1}$)[47]. We flowed a range of PS suspensions in $10^1$–$10^4$ MPs L$^{-1}$ concentrations through AgF@AgM@C10 and collected SERS surveys at each concentration. We increased the number of spectra input into SpecATNet gradually to find the minimum required for a high-accuracy predictions (Fig. 5a). This capability of

SpecATNet to handle varying numbers of spectra and process them concurrently is an advantage of self-attention-based architectures over common CNNs that process single spectrum inputs[16,17]. Lower concentrations of PS MPs require more spectra to achieve high accuracy. It is also important to note the tradeoff between computational load and accuracy because processing more spectra to obtain a high level of accuracy requires more resources. Only 3 to 4 spectra are required to identify PS in a $10^4$ MPs L$^{-1}$ suspension, in contrast to 10 required spectra for $10^2$ MPs L$^{-1}$. In the case of lower concentrations such as 50 to 10 MPs L$^{-1}$, more than 15 spectra are required to achieve high accuracy. We also collected SERS spectra from a single PS particle– 6–8 spectra are required to identify the single particle (Fig. 5b). Importantly, all leading SpecATNet indicators, such as precision, recall and F1 increase with more input spectra, reaching a plateau at 10 spectra, with an average accuracy of 96%. (Fig. 5c).

Next, we increased the multiplexity of the experiments by examining 2-component MP mixtures (Suppl. Fig. 25 and Suppl. Table 6). All permutations of the 5-MPs in 2-component mixtures are plotted as confusion matrices to identify cases where the system mislabels the two polymer classes (Fig. 5d). SpecATNet has an accuracy of >95% with these samples; however it is less accurate with polymers that have

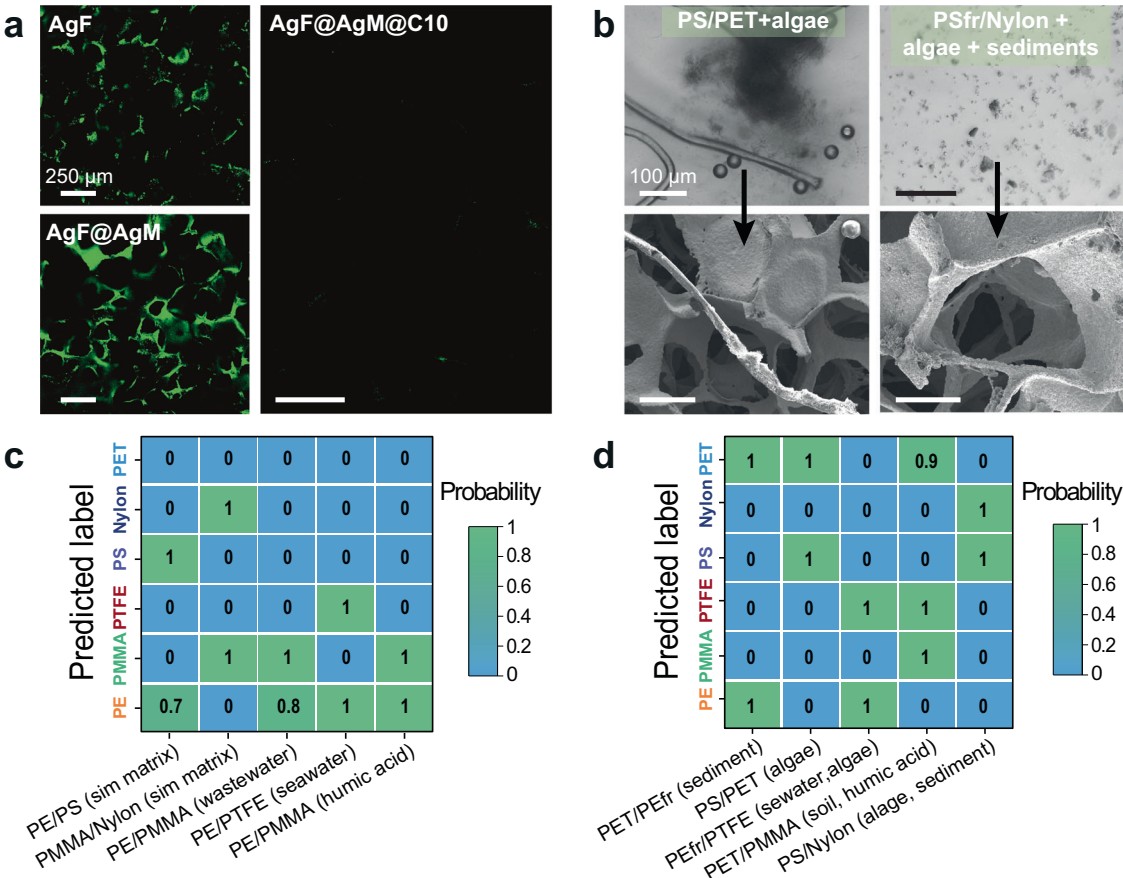

**Fig. 6 | Analysis of realistic samples containing degraded MPs in different matrices. a** Fluorescence microscopy images of AgF, AgF@AgM, and AgF@AgM@C10 following protein adhesion tests using BSA-FITC, scalebar is 250 μm. **b** Microscope images of samples containing algae (3 mg L⁻¹), soil (1 g L⁻¹) (as an example of environmental matrix) and degraded MPs and SEM images of AgF@AgM@C10 after flowing MPs in environmental matrices. Scalebar is 100 μm. **c** A confusion matrix visualizing the performance of SpecATNet for the analysis of multiplexed environmental samples, wastewater, seawater and samples containing humic acid. **d** A confusion matrix visualizing the performance of SpecATNet for the analysis of the multiplexed samples, containing MPs (PEfr refers to PE fragments), algae, soil, sediments, and humic acid.

similar spectral features (Suppl. Fig. 27b). Thus the average accuracy of the two-component mixture was 93.7% (Suppl. Fig. 27b). Despite the different structures and sizes of the MP and differences in SERS EFs (Suppl. Figs. 17 and 19), SpecATNet is capable of identifying each type of MP independent of size and structure. Increasing the multiplexity of the samples to up to 5-component MP mixtures (Fig. 5e) resulted in an average accuracy of 93.1% (Suppl. Fig. 27b). The lower accuracy with increasing multiplexity is expected as the number of possible classes increases. The importance of self-attention in the analysis of multi-component samples is visualized in Suppl. Fig. 27c. Applying self-attention to complex MP samples increased prediction accuracy by ≈10% due to better identification of distinct spectral patterns. No spectroscopic studies have successfully analyzed multi-MP mixtures, especially at low concentrations without PCPT methods. For instance, T. Serizawa et al. analyzed multiple polymer species by peptide-based fluorescence sensing combined with linear discriminant analysis[48]. The main limitation was its applicability only to uncommon water-soluble polymers in high concentrations (10 mg L⁻¹). In contrast, the AgF@AgM@C10 can detect PS at ≈6000× lower concentration (0.0015 mg L⁻¹).

**In-situ sensing of MPs in unprocessed environmental samples**
MPs found in the environment begin as components of plastic products that are increasingly pulverized into smaller pieces with irregular shapes. Physical and biological processes can also modify their surface chemistry[49]. To generate such realistic MPs, we artificially aged

commercial polymers using a photo-Fenton procedure using oxidants and UV light[50] (Suppl. Fig. 28). Besides beads and fibers, MP shape can vary widely among fragments, films, and foams (Suppl. Table 6). The size and shape of the MPs changed significantly (Suppl. Fig. 28). Degrading the MPs had minimal effect on the SERS EFs of the polymers (Suppl. Fig. 29). Environmental MPs samples can originate from different sources; therefore the degraded MPs were detected in different matrices, namely, simulated matrix (common inorganic (0.5 wt.%, NaCl)[51] and organic (0.3 mg mL⁻¹ BSA)[52] materials, humic acid (10 mg L⁻¹)[53], algae (3 mg L⁻¹, Suppl. Fig. 30), reference groundwater sample ERM-CA616 from Belgium (Suppl. Table 7), synthetic seawater, reference marine sediment enriched by organochlorine pesticides (CRM 7304-a, Japan), and reference soil enriched with Cr (CRM041).

Firstly, a mixture of PS, BSA and NaCl was flowed through AgF@AgM@C10, SiO₂, Klarite and flat AgM substrates (Suppl. Fig. 31). The porous structure of the AgF@AgM@C10 sample trapped the MPs, whereas the other substrates did not. If matrix components (BSA and NaCl) are attached to Ag surface, then their SERS peaks would overlap with the peaks of the MPs and create more complex SERS spectra (Suppl. Fig. 32). To examine the antifouling effect, we flowed fluorescein-conjugated BSA (BSA-FITC) on the AgF, AgF@AgM, AgF@AgM@C10, and AgF@AgMCOOH samples and then imaged them using fluorescence microscopy (Fig. 6a and Suppl. Fig. 33). Unprotected AgF adsorbed BSA-FITC and generated strong fluorescence. Fluorescence on AgF@AgM@C10 was suppressed, indicating little BSA fouling. SEM and fluorescence imaging prove that the

4-decylphenyl coating of the SERS sensor preferentially attracts MPs and rejects matrix components, likely because the absolute magnitude of the capillary forces imposed on molecules is small compared to MPs.

Next, we examined the AgF@AgM@C10 samples using the photocatalytically aged MPs with different sizes and shapes in the presence of simulated matrix, humic acid, wastewater, seawater, algae, soil, and sediments and analyzed with SERS and then processed through SpecATNet (Fig. 6b–d). These samples have the highest level of complexity due to their diverse size, shape, and chemical composition. The heterogeneity of the matrix components further complicates analysis (Fig. 6b). The Raman signals of MPs and the environmental matrices overlap with each other (Suppl. Fig. 32). However, all samples tested were identified with an average accuracy of 94% (Fig. 6c, d and Suppl. Fig. 34a). Two 5-component environmental mixtures (Fig. 6c, d) were identified correctly; for example, the sample containing aged PET and PS in the presence of algae (Fig. 6d) was identified with 100% accuracy. The sample containing aged PS fragments from foam and Nylon (Fig. 6d) was identified with 97% accuracy. To test the performance of SpecATNet with a negative control experiment, AgF@AgM@C10 was treated with BSA/NaCl, wastewater, sediments, soil, algae, and seawater without any MPs. According to Suppl. Fig. 34b, SpecATNet has low false-positive performance.

The importance of processing multiple spectra simultaneously with the self-attention architecture can be understood by evaluating how the number of spectra impacts accuracy (Suppl. Fig. 35). For example, if SpecATNet is served SERS data in batches with <5 spectra, the accuracy drops to ≈80-90% (Suppl. Fig. 35a). With every additional spectrum input into SpecATNet, the NN learns proper averaging and reaches maximum accuracy with a minimum 10 spectra. The other types of vibrational spectroscopy methods—including methods assisted by NNs— have not demonstrated similar performance on MPs in complex matrices, especially without PCPT methods. More specifically, AgF@AgM@C10 coupled with SpecATNet was compared with other recently reported SERS platforms (Suppl. Table 5). AgF@AgM@C10 outperforms noble metal NP sensors[54,55] and commercial Klarite[12] in its ability to sense diverse MPs (structure, size, shape), complexity of samples, and accuracy at realistic low concentrations. In a recent report, a SERS substrate based on porous paper demonstrated the simultaneous detection of PS and PE at high concentrations (down to $10$ mg L$^{-1}$ or $\approx 10^{10}$ MPs L$^{-1}$)[56]. They used logistic regression to interpret spectra, but binary classification falls short for complex samples, as it does not account for nonlinear relationships between the independent variables, such as multiplexity of data. In contrast, we analyzed 6 types of MPs with different sizes and shapes while suspended in various organic matrices and interference agents. Combining hierarchical macroporous-mesoporous metal foams with the self-attention-based CNN SpecATNet allowed us to analyze down to 0.00015 mg L$^{-1}$ or 10 MPs L$^{-1}$, allowing our workflow to access concentrations found in marine and freshwater samples without requiring PCPT methods.

The main alternative method to detect multiplexed samples is pyr-GCMS, which requires PCPT methods[57]. Timelines were proposed for both methods to compare the throughput of pyr-GCMS versus SpecATNet coupled with AgF@AgM@C10 (see Suppl. Fig. 36 and Suppl. Note 4). The pyr-GCMS measurement itself is relatively fast, but the PCPT methods and spectral analysis require the most time, resulting in ≈0.1 predictions per h (Suppl. Table 1). In contrast, SERS-SpecATNet can generate 2–4 predictions per h, mainly by omitting PCPT methods, resulting in a 20 times faster workflow.

Previous reports use multi-class classification to recognize different MPs, so we initially used accuracy to compare their performance with SpecATNet (Suppl. Fig. 27). However, the accuracy of multi-label classification models does not account for the specific nature of multi-label problems, where samples can have many labels

simultaneously[40,45]. Precision and recall are more relevant in this context because they provide deeper insights into model performance (Suppl. Fig. 37). The level of precision for SpecATNet is from 82 to 98%, indicating that it makes many correct positive classifications and a low number of incorrect positive classifications. Recall measures the ability of SpecATNet to detect only positive samples. The recall of SpecATNet is 62-95%. Precision-recall curves (PRC) show the precision and recall values at various thresholds (Suppl. Fig. 38a). High precision indicates that positive predictions are reliable, while high recall indicates that the model can capture most of the positive samples in the dataset. We further generated the receiver operating characteristic curve (ROC) by plotting the sensitivity against the false-positive rate (Suppl. Fig. 38b). The ROC curve shows sensitivities and specificities significantly higher than random classification. By varying the classification threshold, it is possible to examine trade-offs between sensitivity (true positive rate) and specificity (true negative rate). A noninformative threshold 0.5 was set here since no assumption about price of type I/type II errors was made. However, the threshold can be chosen based on environmental threat, location of the sample probe, regulatory requirements etc. Of the 6 polymers examined, PS and PET have the highest area under PRC, representing both high recall and high precision, and there is a tendency in MP recognition reliability: PET > PS > PMMA > PE > PTFE>nylon. The smallest MPs are less precisely recognized versus micron-sized MPs due to their lower probability of being measured in the SERS survey. The smaller MPs may also hide underneath and overlap with the larger MPs. Finally, the balanced parameter F1 score is used to compare SpecATNet with other NN-based systems. The F1 scores for tested MP are within 85–96%, within the same level as other NN systems used for binary classification of single MPs with FTIR[45] or Raman[40].

We compared averaged F1 scores for all MPs with the F1 score obtained from other common machine-learning methods, such as a logistic regression model, decision tree, and support vector machine (SVM) using our dataset (Suppl. Fig. 39). The linear logistic regression has the lowest F1 score (69.2%) due to its inability to capture complex, nonlinear relationships in MP mixtures. The decision tree method has a higher F1 score (74.6%) due to nonlinearity; however, it could have a limited ability to capture and represent complex or intricate relationships within the data, leading to suboptimal predictive performance. The nonlinear SVM model is capable of recognizing complex mixtures with a F1 score of 82.7%. However, the ability of SpecATNet to combine information from multiple spectra gives the highest F1 score (89.3%) compared to other prediction approaches.

Despite the high performance of the developed sensing procedure, there are some features that could be improved with further iterations of the SERS sensor, NN and analytical setup. They are: (1) the micropore size of the Ag foam used in this work determines the maximum size of MPs −262 μm. However, macroporous templates with larger pore sizes or more complex networks could be generated using additive manufacturing[58]. And much larger MPs are commonly determined by visual methods[59,60]. (2) Ag does not have an indefinite shelf-life. Despite preliminary results show the accuracy in identifying MP mixtures remains consistent after one month of air storage for PTFE/PE and PMMA/PS (Suppl. Fig. 40), additional stability studies are needed. Using mesoporous Au would improve stability and enable measurements at near-infrared wavelengths. (3) Small Raman spectrometers can be made inexpensively using off-the-shelf parts (<3600 $ according to OpenRAMAN)[61] but more work must be done to design complete optical setups that are easier to deploy even for the most resource-limited labs. (4) Upgrading of SpecATNet as with any other NNs requires the collection of a large dataset. For example, a minimum ≈4000 spectra are required to introduce additional MPs with different chemical structures. Collecting a large and diverse dataset of spectral

data for training CNNs can be challenging, while limited data can lead to overfitting and poor generalization to unseen spectra.

In conclusion, we described a SERS sensing platform capable of identifying MPs in multi-component samples without onerous PCPT protocols. The SERS substrate uses physicochemical interactions from the macroporous-mesoporous structure and the hydrophobic 4-decylphenyl protecting layer to favor the trapping of MPs over small water-soluble molecules while stabilizing the metal surface from oxidation. The macroporous-mesoporous metal foam also enhances light coupling and formation of EM hotspots that can generate strong SERS scattering from MPs trapped inside the porous network. The economic cost of these macroporous networks are ≈10−20× lower than commercial SERS substrates and may be further optimized for MP sensing by adapting the deposition method to work on cheaper foam substrates such as nickel, conductive polymers or custom 3D printed substrates that could also enhance both light absorbance and MP trapping (Suppl. Fig. 41, Suppl. Table 8, and Suppl. Note 5).

To demultiplex information about MP chemical structure from the dataset, we designed a NN called SpecATNet to analyze patterns in the SERS survey spectra and identify MPs. SpecATNet borrows image-like processing algorithms from CNNs that use pixels as input. In our experiments, inputs are SERS spectra collected during the survey of unknown samples. The self-attention mechanism, borrowed from NLP, independently assigns weights to each spectrum and then averages them to make a final prediction about the presence and structure of MP. In principle, self-attention-based CNNs can be used with any source of spectroscopic data−even multiple types of spectroscopy−to analyze multi-component samples to increase accuracy. The porous metal substrates have sufficient affinity for MPs. The tortuous network of macropores and mesopores traps MPs from a flowing solution and avoids onerous and time-consuming PCPT methods. The metal surface surrounds the MPs, enabling the plasmon resonance of the metal to generate strong SERS signals and convert that data into machine-readable chemical information. The long-term goal of this work is to enable measurements of MPs directly at pollution sources, which can be analyzed on-site with a local NN on a PC or uploaded to the cloud for analysis. Researchers could further upgrade SpecATNet by training it to identify other hydrophobic MPs, common organic contaminants, polycyclic aromatic hydrocarbons, drugs, and dyes. By merging porous metal structures with self-attention-based NNs into a PCPT-free sensing workflow, people can rapidly identify MPs with varying structures, sizes, and levels of degradation.

## Methods
### Materials
A 1-mm thick sheet of silver foam with an average pore diameter of 0.16 mm was purchased from Axel. Silver nitrate (ACS reagent, ≥99.0%), 4-decylaniline (97%), p-toluenesulfonic acid monohydrate, bovine serum albumin (≥98%), ethanol, wastewater (ERM-CA616), soil (Chromium VI – Soil, RTC CRM041-030), synthetic seawater (SSWS30) and tetrahydrofuran (≥99.9%) were purchased from Sigma Aldrich and used without additional purification. Tert-butyl nitrite (>90.0%) was purchased from TCI Chemicals. Humic acid (practical grade) was purchased from Fujifilm Wako. Marine sediments with polychlorinated biphenyls and organochlorine pesticides (NMIJ CRM-7304-a) was purchased from the National Metrology Institute of Japan. Polystyrene-$b$-poly(ethylene oxide) diblock copolymers with $M_w$ 18,000 g mol$^{-1}$(PS) and $M_w$ 7500 g mol$^{-1}$(PEO) molecular weight subunits were acquired from Polymer Source (PS$_{18000}$-b-PEO$_{7500}$). Commercial SERS surfaces were used in this study: Klarite® 313 (Renishaw Diagnostics, Ltd). PS beads (30 μm) and free-flowing PTFE beads (1 μm) were purchased from Sigma Aldrich. Nylon−12 beads (5 μm) were purchased from Toray Plastics. PMMA beads (27−45 μm) and PE beads (10−90 μm) were purchased from Cosperic. PET fiber and expanded PS foam (polymer kit 1.0) were purchased from Hawaii Pacific University

Center for Marine Debris Research. For the flow experiments, we used μ-Slide I Luer cell with a channel height of 0.8 mm, width of 0.5 mm, and length of 50 mm (Ibidi). Spirulina powder was purchased from Tree of Life (Japan). To convert MP concentration from mg L$^{-1}$ to MPs L$^{-1}$, the size distribution was considered (Suppl. Fig. 17) following Eq. 1, according to ref. 62:

$$C_{\text{MPs per L}} = \frac{C_{\text{mg per L}} * 10^9}{\left(\frac{\pi}{6}\right) \times \rho \times \left(\sum D_n \times P_n\right)^3},\tag{1}$$

Where $\rho$ is the density of polymer in g cm$^{-1}$, $D_n$ is the diameter of the MP bead in μm, and $P_n$ is the percent content of this fraction. Recalculated values are given in Suppl. Table 4.

**Preparation of degraded MP.** Photo-Fenton oxidation of MPs was carried out in a photoreactor equipped with an EvoluChem PhotoRedOx Box (HepatoChem, USA). The average light intensity was ≈1.0 mW cm$^{-2}$ (at 365 nm). 0.40 mL of 20 mM FeCl$_3$ and 0.45 mL of 30% H$_2$O$_2$ were added to 19 mL water containing 0.15 mg L$^{-1}$ MPs under vigorous stirring. The irradiation time was different for each polymer: PE (3 h), PS (24 h), Nylon (24 h), PTFE (72 h), PMMA (24 h), PET (24 h), 3 to 5 mm diameter PS pieces were grated with a nickel foam (pore size ≈ 180 μm) to obtain microsized pieces, PE film pieces (preliminary ground to ≈150-500 μm). After irradiation, the MPs were centrifuged (7000×$g$, 15 min) two times: washing with water in the first step, then methanol in the second step, and finally dispersing the MPs in 20 mL of water (Suppl. Fig. 28).

**Preparation of MP solutions that mimic environmental samples.** Real environmental samples contain protein and salt. To mimic these samples, we mixed 1.6 mL of bovine serum albumin (BSA; 30 mg L$^{-1}$) and 1.6 mL of 0.5% NaCl with 0.4 mL of an MP solution (0.15 mg L$^{-1}$). This mixture was sonicated in a water bath for 1 min to homogenize the sample and then used for analysis. The algae sample was grown in Guillard's (F/2) Marine Water Enrichment Solution for 7 days, washed by water via centrifugation (3000×$g$, 3 min, 3 cycles), and used in 3 mg L$^{-1}$ concentration (dried algae). Humic acid was used in 10 mg L$^{-1}$ according to ref. 53. Soil and sediments were used in 1 g L$^{-1}$ concentrations. The following ratios were used to prepare the complex mixtures: algae/seawater – 1:1; humic acid/soil (Cr) – 1:1; algae/sediments – 1:1.

**Preparation of AgF@AgM.** The procedure was adapted from an earlier report[63]. 5 mg of PS$_{18,000}$-$b$-PEO$_{7,500}$ block copolymer was mixed in 1 mL of THF and stirred in a water bath at 40 °C for 1 h to fully dissolve the polymer. Micelle formation was previously examined with TEM[63]. Then 0.5 mL of EtOH, 0.5 mL of aqueous AgNO$_3$ (40 mM), 0.5 mL of HNO$_3$ (0.5 M), and 3 mL of dionized water were added to the polymer solution in this order, and then the solution was maintained at 0 °C in an ice water bath. All electrochemistry experiments were performed using an electrochemical workstation (CH Instruments 660E). Immediately before the electrodeposition experiments, the Ag foam (0.5 × 0.7 cm$^2$) was placed in a solution of 3.5 M HNO$_3$ for 15 min to remove the surface oxide layer and then briefly washed with water. The electrodeposition process was performed using a 3-electrode setup that included a working electrode of Ag foam, a reference electrode of Ag/AgCl, and a counter electrode composed of Pt wire. During the entire electrodeposition experiment, the 3-electrode setup was kept ice-cold (0 °C). Electrodeposition proceeded at -0.25 V for 1000 s. After the deposition, the AgF@AgM substrates were rinsed with THF overnight to remove residual PS-b-PEO and other solvents. Suppl. Fig. 5 confirms the absence of PS-b-PEO-related peaks after washing. After the removal of micelles, well-defined porous structures can be observed in SEM. The deposition temperature was held at 0 °C to prevent the Ag$^+$ ions from spontaneously reducing in solution without

applied potential. SEM images indicate that the AgM coats 95% of the AgF surface. Some patches of missing pores are likely due to the non-uniform potential caused by the convoluted surface during electrodeposition.

**Preparation of AgF@AgM@C10.** 4-decylbenzenediazonium tosylate (ADT-C10) and 4-carboxybenzenediazonium tosylate (ADT-COOH) were prepared according to an earlier report[64]. The AgF@AgM was immersed in a freshly prepared MeOH:$H_2O$ (3:2) solution containing 1 mM ADT-C10 for 1 h. The freshly passivated AgF@AgM@C10 substrate was washed with water (2 times) and EtOH (2 times).

**Deposition of MPs on AgF@AgM@C10.** In the typical experiments, the AgF@AgM@C10 (0.5×0.5 cm$^2$) substrate was placed inside the μ-Slide I Luer cell (Ibidi, USA), which is typically used for flow experiments. 100 mL of the MP solution was cycled through the cell using a peristaltic pump at 5 mL min$^{-1}$. To determine the sensitivity of AgF@AgM@C10 to MP concentration, different PS suspensions containing $10^1$ to $10^4$ particles per liter (MPs L$^{-1}$) were cycled through the cell at 5 mL min$^{-1}$. For the experiments with low concentrations of PS (<100 MPs L$^{-1}$), the circulation time was 1 h. The MP suspension was initially prepared by weight concentrations (mg L$^{-1}$) and further recalculated as particles per liter (MPs L$^{-1}$) considering the size distribution based on ref. 62. For example, the 15 mg L$^{-1}$ sample was initially prepared by adding 3 mg of PS powder to 200 mL of solution (198 mL of water and 2 mL of EtOH) and then sonicated for 30 min to prevent adsorption to the mixing vessel. Lower-concentration samples were prepared by diluting the 15 mg L$^{-1}$ solution. For example, the 0.15 mg L$^{-1}$ PS suspension was prepared by diluting the 15 mg L$^{-1}$ suspension ×100 to generate a final volume of 200 mL. 100 mL of this solution was flowed over the metal foam sensors via recirculation. Further dilutions were performed in a similar manner. Afterward, AgF@AgM@C10 was removed, dried in air, and analyzed.

**Optical simulations of mesoporous Ag films and macroporous Ag foams.** A 500 nm × 500 nm section of the mesoporous Ag surface was taken from a SEM image and converted into a 2D mesh by representing the pores and surrounding environment as air (refractive index, RI = 1) and the Ag metal using optical constants described by McPeak. The mesh was imported in an EM modeler (Lumerical-Ansys) as a 50-nm thick mesoporous film and placed on a thick slab of Ag. The mesoporous film was illuminated with a plane wave, and reflectance was monitored from a location 1000 nm above and parallel to the surface of the mesoporous metal film, while the local intensity of the EM field was monitored 3 nm above the film. Ag foams measured by X-ray CT were converted into STL files. A 478 μm × 560 μm × 450 μm section of the Ag foam was input into the EM modeler as a 3D mesh object. The surface charge distribution of the foam surface was modeled to observe how light coupled to the surface and formed gradients that suggest the presence of surface plasmon polariton modes.

**Raman spectroscopy measurements.** Raman spectra of the MPs were collected on an old JASCO NRS3100 spectrometer (produced 2005) with 532 nm laser excitation (laser power at sample, 6 mW) and a resolution of 2 cm$^{-1}$ (600 g mm$^{-1}$ grating) spanning 1400 to 400 cm$^{-1}$. SERS survey spectra are measured using mapping mode with a ×20 objective (Olympus UMPlanFl ×20/0.46 NA BD objective, 3.0 mm working distance). In mapping mode, the objective is parked over the substrate and then scanned over various areas from 0.1 × 0.1 mm$^2$ to 0.3 × 0.5 mm$^2$ with a collection time of 10 s per point. For the calculations of EFs, a minimum of three values have been measured and intensities were used with standard deviation (SD) calculated from $N$ spectra according to the relation SD $= \sqrt{\sum \left| \frac{x - \bar{x}}{N} \right|}$, where $x$ is the intensity of the Raman signal, $\bar{x}$ is the mean of the intensities, and $N$ is the number of spectra used.

Raman spectra on gold-based Klarite substrates were collected on a Renishaw inVia Reflex Raman Microscope with 785 nm laser excitation (laser power at sample, 20 mW) and a 1200 g mm$^{-1}$ grating. The spectra were acquired using a ×20 objective and a 10 s collection time. 3D mapping measurements were performed on inVia™ confocal Raman microscope (Renishaw, United Kingdom) with 532 nm (6 mW, 10 s for each exposure). The map of one selected area (584 μm × 584 μm × 60 μm) was acquired with a resolution of 8 μm × 8 μm × 20 μm in $X$, $Y$, and $Z$.

**EF calculation.** Calculation of the SERS EF was performed according to the standard relation: EF $= \frac{I_{SERS}/C_{SERS}}{I_{RS}/C_{RS}}$, where the $I_{SERS}$ and $I_{RS}$ represent the Raman scattering intensities on SERS-active and reference silicon surfaces, and $C_{SERS}$ and $C_{RS}$ are the corresponding concentrations of R6G.

For measuring of $I_{RS}$ on silicon $10^{-1}$ M of R6G aqueous solution was used. To accurately estimate the number of molecules excited by the confocal microscope, a ×20 objective lens with a 1.1 mm beam waist was illuminated through a container with a depth of 3 mm. Assuming the Gaussian beam excites a volume that is roughly the shape of a cylinder, we can estimate the volume excited by the beam using the Eq. 2:

$$V = 0.5 \times D \times \pi \times h \qquad (2)$$

where $D$ is the beam waist, and $h$ is the height of the container. By multiplying the concentration of the solution (in molecules μm$^{-2}$) by the beam volume, we can estimate the number of molecules excited in the Raman reference measurement ($C_{RS}$).

**μ-CT 3D images.** The 3D models of the Ag foam material were obtained with an X-ray μCT system SKYSCAN1275 (Bruker, USA) with a current of 85 kV and a voltage of 114 mA. The rotation step was 0.2°.

**Electrocatalytically active surface measurements.** Electrocatalytically active surface areas (ECSAs) of the samples were obtained in $N_2$-saturated 0.5 M $H_2SO_4$ with a scan rate of 50 mV s$^{-1}$ based on the peaks of reduction of Au/Ag oxides in the potential range of -0.4 to 0.4 V vs. Ag/AgCl from cyclic voltammetry[65,66]. The electrochemically active surface areas (ECSAs) were calculated using the charge associated with the reduction of oxide by integration by Eq. 3:

$$ECSA = \frac{Q}{Q_{ref}} \qquad (3)$$

where, $Q$ is specific capacitance of the electrode of electrode in cm$^2$ per scan rate V s$^{-1}$ and $Q_{ref}$ is reference specific capacitance of gold.

**Scanning electron microscopy.** The morphology of the samples were characterized using a Hitachi SU-8000 field-emission scanning electron microscope at an accelerating voltage of 10 kV.

**Wettability measurements.** The surface energies of the Ag substrates were measured using the OWRK model via the contact angles of water and ethylene glycol. All measurements were performed at room temperature on a VCA Optima-XE at ten positions with a drop volume of 2 μL.

**X-ray photoelectron spectroscopy.** XPS was performed using a Thermo Fisher Scientific XPS NEXSA spectrometer with a monochromated Al K$_\alpha$ X-ray source operating at 1486.6 eV. XPS survey measurements used a pass energy of 200 eV and an energy resolution of 1 eV. High-resolution XPS spectra were collected using a pass energy of 50 eV and an energy resolution of 0.1 eV. The analyzed area was 200 μm$^2$, and a flood gun was used for charge compensation.

**Optical measurements.** The absorbance spectra of AgF and AgF@AgM were collected using a JASCO V-770 spectrophotometer.

**Fluorescence microscopy.** To test antifouling properties the AgF, AgF@AgM, and AgF@AgM@C10 were each placed separately inside a Luer cell, then a 1 mg mL$^{-1}$ BSA−FITC solution was cycled through the cell at 5 mL min$^{-1}$ for 10 min. Afterward, the samples were removed from the cell and thoroughly rinsed with deionized water. Fluorescence imaging was performed on a confocal microscope (Leica TCS SP5) and an appropriate light filter (450 nm excitation; 550 nm emission).

**Adsorption capacity measurements.** For adsorption capacity measurements AgF, AgF@AgM, AgF@AgM@C10 and AgF@AgM@COOH substrates were placed inside the µ-Slide I Luer cell, typically used for flow applications. 100 mL of the PS suspension (1 mg L$^{-1}$) was cycled through the cell at 5 mL min$^{-1}$ using a peristaltic pump. The adsorption capacity was calculated as a difference in mass before and after exposure to PS suspension adsorption divided by substrate mass.

**Calculation of SD for wettability, surface free energy, and adsorption capacity measurement.** For the calculations, a minimum of three values have been measured, and they were used with SD calculated from $N$ measurements according to relation $SD = \sqrt{\sum \left|\frac{x-\bar{x}}{N}\right|}$, where $x$ is the measured value, $\bar{x}$ is the mean of the measured values, and $N$ is the number of measurements used.

**SpecATNet architecture.** SpecATNet is a combination of commonly used CNNs and Transformers. A DenseNet-like architecture was used as the CNN component due to its successful application in the recognition of Raman spectra[67,68], providing good performance and fast convergence while being relatively simple. DenseNet is a type of deep-learning NN where sequences of input data are processed stepwise. The convolutional backbone size was restricted to a single DenseNet block. The network parameters were optimized by grid search with a HyperBand pruning strategy[69].

The SERS survey dataset from each sample has been collected in a mapping mode consisting of points ($S_1$, $S_2$... $S_n$) over the scanned area. DenseNet served as an encoder for SERS data. Each spectrum ($S_1$, $S_2$... $S_n$) was transformed into a hidden representation ($Z_1$, $Z_2$... $Z_n$) to reduce the sequence dimension. Further, SpecATNet combines representations of all spectra; that is, it reduces the sequence dimension followed by weighting the reduced representation to generate final predictions with the fully connected layer. For the weighting, the self-attention layer was implemented after DenseNet. Attention dynamically adjusts the weights of individual elements in a sequence set, where the importance of each element is determined by its relationship to the others. After passing the self-attention layer, the data is reduced by simply averaging along the sequence axis. The output of the convolutional backbone for every input spectrum is reduced to the size 64 by a single fully connected layer and is passed to a scaled dot-product attention[18] with a single head. The output of the attention layer is averaged and directly connected to the last fully-connected layer, which outputs logits (i.e. predictions).

**Dataset and training**

The list of collected spectra is in Suppl. Table 6 and available from ref. 70.

**Data pre-processing.** Raw Raman spectra were background-subtracted with the arPLS algorithm[71], then cosmic spikes were removed by detecting them based on derivative value and removing them via interpolation. Finally, spectra were normalized by subtracting the mean and dividing by the standard deviation. Since multiple spectra are available for a single sample, splitting them into training and validation sets is unnecessary. Instead, the splitting should be

done on a sample level, with all the spectra of a sample put in the same split. Additionally, it was necessary to generate sequences of spectra of the desired length using the following algorithm:

Given: samples $Sp_1$...$Sp_N$, set of spectra for every sample, sequence length L, batch size B
1. Randomly choose a single sample $Sp_i$ from available samples.
2. Randomly sample (with replacement) L spectra from the set of $Sp_i$ spectra
3. Repeat steps 1 and 2 until B sequences are collected

In this way, we inherently guarantee that the probability of being selected is the same for each sample, even though different numbers of spectra are available for different samples. We also use sampling with replacement, which ensures that it is always possible to get a sequence of the desired length even if there is an insufficient number of spectra.

**Data balancing.** Although the sampling algorithm balances the contributions of individual samples, the relative abundance of polymers differs between samples, which might to data imbalance. The multi-label classification can be considered as multiple binary classifiers with shared weights, therefore, every polymer would have approximately the same abundance (0.5 in the best case) to guarantee that all of them will be equally important for the NN. The most straightforward strategy is random oversampling, i.e., repeating minority class to balance the data. Despite being trivial for multi-class classification, searching for oversampling coefficients becomes an NP-complete task of integer programming. The task is formulated as:

Given the binary matrix $A$ (with dimension $N_{samples}$ x $N_{clas}$), containing multi-hot encoded labels for every sample in the dataset, find the integer vector $x$, such that $A^T x / N_{samples} \approx [0.5, 0.5, ...]$, $\forall x_i \geq 1$.

This task has been reformulated as optimization and solved using SCIP solver[72]. Precisely, the objective is: $\min \|A^T x - b\| + 0.1 \text{maximum}(x)$ where $b = [0.5, 0.5, ...] * N_{clas}$ under constraint $\forall x_i \geq 1 \land x_i \leq 5$ The constraint and penalty on maximum value of $x$ is added to force the solver to select dense solution (near all samples are oversamples with small coefficient) rather then sparse solution (only several sample are oversampled with high coefficient). After obtaining the solution, fictitious samples were generated by copying existing ones and adding them to the dataset.

**NN training.** The data was split into training and test sets in a 10-fold cross-validation manner to train the network. The network was trained using sharpness-aware minimization[73] with a stochastic optimization method called AdamW[74] as the base optimizer (learning rate = $1 \times 10^{-3}$, weight decay = 0.1). The data sequence length was randomly chosen from 4 to 32 for every training iteration. The network was always trained for one epoch; any further training was found to destabilize validation metrics. Increasing regularization does not influence validation metrics destabilization.

**Ablation study**

We studied the influence of self-attention block and sharpness-aware minimization on network performance. For the former case, the self-attention was removed so that the averaging was performed directly on every spectrum embedding. This led to $a \approx 3\%$ decrease in the final cross-validated accuracy. Replacing the sharpness-aware optimizer with a simple AdamW led to $a \approx 1\%$ decrease in accuracy.

**Machine learning methods for comparison**

For comparison, we used classical algorithms such as logistic regression (LogReg), SVM, and decision trees (DT) to classify individual spectra. We transformed it into multiple binary classification tasks using the MultiOutputClassifier from scikit-learn to handle multi-label classification. We pre-processed the spectral data by normalizing and

reducing its dimensionality to 32 components through PCA, optimized via grid search. The evaluation was performed using a 10-fold cross-validation approach.

## Reporting summary

Further information on research design is available in the Nature Portfolio Reporting Summary linked to this article.

## Data availability

The data that support the findings of this study are available from Kaggle[70], Zenodo[75], and from the corresponding authors upon request.

## Code availability

Codes are deposited at Zenodo[75] and are available from the corresponding authors upon request.

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

## Acknowledgements

The authors gratefully acknowledge the assistance of Dr. Lambard Guillaume for discussion and Mr. Shota Mitani for X-ray CT observations. This research was supported by the JST-ERATO Yamauchi Materials SpaceTectonics Project (JPMJER2003) and the Japan Society for the Promotion of Science (JSPS) Grants-in-Aid for Scientific Research Kakenhi Program (20K05453). J.H. acknowledges the LaSensA project under the M-ERA.NET scheme funded by the Research Council of

Lithuania (LMTLT, agreement no. S-M-ERA.NET-21–2), Saxon State Ministry for Science, Culture and Tourism (Germany), and the National Science Centre (Poland). OG acknowledges JSPS Postdoctoral Fellowship, and the Korea Institute of Industrial Technology (KITECH, JE210028). PS and OG acknowledge RSF 23–73-00117 (the design, preparation and characterization of plasmonic nanostructure). A part of this work was supported by the Advanced Research Infrastructure for Materials and Nanotechnology in Japan (ARIM) of the Ministry of Education, Culture, Sports, Science and Technology (MEXT) proposal number JPMXP1224NM5002.

## Author contributions

O.G. developed the concept, performed most of the experimental work, collected data, and wrote the manuscript. A.T. developed and trained CNN algorithm, and wrote the manuscript. Y.K. performed electrochemical experiments. P.P. assisted in concept development. M.K. and A.S. performed X-ray CT experiments and described them. L.K.Sh. contributed to Raman measurements. J.H. developed the concept, performed EM simulations, wrote the manuscript, and supervised the project. Y.Y. revised the manuscript, contributed to the discussion, and supervised the project.

## Competing interests

The authors declare no competing interests.
