## [Peer Review File · Nature Communications]

Pretreatment-free SERS sensing of microplastics using a self-attention-based neural network on hierarchically porous Ag foamREVIEWER COMMENTS

Reviewer #1 (Remarks to the Author):

Rapid identification of microplastics have become an urgent need in the field of microplastic pollution, but most of the methods now used are time-consuming and needing independent separation or pre-treatment protocols. This study established a method using SERS sensing coupled with self-attention-based neural networks to identify microplastics, which has a potential to be applied in some other fields. They could analyze complex samples with multiple MPs simultaneously and identify the chemical composition of MPs rapidly with a high accuracy. Generally, this is a useful method with potential application in the field of microplastic identification. However, the authors only analyzed commercial standard quasi-spherical MPs in laboratory, and the effects of this method on other shapes of MPs still need to be checked. Moreover, although they aged the MPs with chemical treatment, the samples are still different from the natural ones. They did not analyze real environmental samples such as water, soil or sediment samples with this method, so this method might be only feasible for some types of commercial MPs. Additional experiments are needed to be conducted to check the effects of this method on environmental samples. There are some issues should be addressed as follows

Major comments

-The main defect of this study is that they do not analyze environmental samples with their method. Additional experiments should be conducted with natural water samples and the detection effect should be presented, otherwise the application of this method will be limited to laboratory analysis of standard samples.

-The selection of five different MPs should be elaborated. 1. We know that some other polymers such as PVC, PP and PET plastics are commonly used and detected in the environment. Why only the five polymer types are chosen? 2. The size selection standard should be illustrated, and could this method detect large MPs (90-500 μm or even 500-5000 μm)? 3. We know that the shapes of MPs vary a lot, including spheres, fragments, films, fibers and foam etc. And in realistic environment, fragments and fibers account for a large proportion. However, only quasi-spherical MPs were tested in this study, some other shapes of MPs should be characterized with this method to verify the effect.

-Generally, the abundance of MPs in freshwater hardly exceeds 100 items per liter. In this study, the flowed suspensions of MPs are unrealistically high (100~10000 particles per liter). How could this method be used without pre-enrichment? The enrichment method of MPs might be developed to facilitate the use of this method.

-Although the commercial microplastics were artificially aged using oxidants and UV light, they could not mimic the environmental microplastics accurately, because there are usually organisms (microorganism, microalgae or fungi) and organic matters attached on the MPs, which might have an influence on their Raman spectrum acquisition. Moreover, why organic material (0.3 mg/mL BSA protein) other than humic acid or fulvic acid was selected? You know that humic acid and fulvic acid are normal organic matters detected in natural water.

-The deficiency or limitation of this method should be presented in the last paragraph of the results and discussion section.

Minor comments

Title, Pay attention to microplastics, this phrase is not highlighted the content of this study, replace or delete

L30, real-time monitoring of MPs is hard to realize by this method, so immense potential is not appropriate.

L36, divided should be degraded
L47, and should be or
L287, delete -
L324, single MP sample
L331, varied should be changed
L334, PS-MPs
Fig.5e is not cited in the text.
L358, Fig.5b should be Fig.5e?
L359, Fig.S25b? please check this.
L366-367, how did you get this value 0.0015 mg/L, could you explain?
L401, add microplastics after PS
L407-408, this study does not analyze really environmental samples either, please revise.
L415, from 80 to 96%
L417, delete within
L431, The conclusion section is a little lengthy
L432-435, These sentences can be deleted
L441-445, This sentence is too long, please revise.
L453-455, it can collect numerous MPs from the environment on-site without onerous separation and analytical methods, this statement is not appropriate, because you don't really check the real effect of this method on-site without separation and analytical treatment.
L468, Why the irradiation time was so different?
L469, minutes should be min
L515, change x with ×
L532-L604, some contents are missing, please check

Reviewer #2 (Remarks to the Author):

In this article, the authors introduce a microplastics (MPs) SERS sensing platform that incorporates designed macroporous-mesoporous silver (Ag) structures with hydrophobic surfaces and neural network algorithms. The unique properties of these Ag structures allow for the direct trapping and separation of microplastics (MPs) from environmental samples. Furthermore, a specialized neural network algorithm, called SpecATNet, can identify various types of MPs (5 in total) in complex multi-component mixtures through multi-label classification of SERS spectra. Overall, the topic of interest in this work is timely and very important and the work was conducted in an organized and comprehensive manner. However, it reads more suitable for a specialized journal as it does not meet the standards for Nature Communications in terms of overall impact.

Specific comments:

(1) For real impact, the speed, throughput and portability would be the key for sensing and classifying MPs. The presented results are very low throughput, are acquired by rather expensive benchtop/large equipment, and it is practically impossible for this technique in its current form to screen large volumes of samples rapidly in a practical setting. Although valuable in their discipline, the results belong to a more specialized journal.

(2) Several clarifications are needed regarding the distribution of the training and test datasets. The authors stated that the datasets were divided in a 10-fold cross-validation manner, and detailed numbers of various sample types are provided in Supplementary Table S6. However, certain sample types, such as PS(AgF@AgM@C10), Nylon(AgF@AgM@C10), and PMMA+PS, have only one sample. This raises the question of how SpecATNet could accurately predict these types when not included in the training set. Could it be that the spectral distributions of these samples were embedded/hidden within the spectra of other MP combinations, allowing for accurate identification even without being

included in the training? Additional analysis and explanations on this matter are needed. Furthermore, Table S6 listed some sample types twice, such as PTFE(AgF@AgM@C10) and PMMA+PTFE, but with different sample counts. The authors should elucidate why these sample types are duplicated and specify the distinctions between each instance.

(3) In the performance analysis of SpecATNet, the authors could include a negative control scenario. Specifically, they could test samples that contain no MPs but include other materials and some potential particles commonly found in realistic environmental water samples—to assess whether the detection of system generates false positives. This additional analysis would offer a more complete understanding of the system's accuracy.

(4) To interpret the predicted scores (decimals from 0 to 1) generated by SpecATNet, a specific threshold is required to determine whether a test sample contains a particular type of MP or not. However, the main text currently lacks this information. The authors should clarify the threshold value used and provide the rationale for its selection. ROC curves are completely missing.

Minor comments:

(1) A formatting issue exists between lines 531 and 663 in the "SpecATNet architecture" subsection of the Methods section, resulting in the missing of several sentences.

(2) In line 359, "Fig. 25b" should be corrected to "Fig. S25b".

(3) The definition of "precision" in Fig. S32 is not correct now.

(4) Some supplementary tables/figures were not cited in the main text, such as Fig. S2, Fig. S11.

(5) The color bars in Fig. 5d, Fig. 5e and Fig. 6b are not matched with the visualizations of their confusion matrixes, where green color means 1 in the confusion matrix but corresponds to low probability in the color bar.

Reviewer #3 (Remarks to the Author):

The article describes a system to detect microplastic in probes using spectroscopies as readout. The claimed novelty is the choice of the substrate and the use of a recently proposed deep learning system. While the article is prepared in a well manner and shows substantial work in detail, a transfer to the problems of the journal's readers seems difficult.

Firstly, I am not sure what the novelty of the approach truly is. Material-preparation is state of the art, spectroscopic read-outs for this kind of problem is state of the art, neural networks are just used and not newly developed. Further, there is a big lack of motivation, it is not described why exactly the used substrates were applied, why the readout of the system is spectroscopic and why not other neural nets are used. Especially in material sciences, simple CNNs have proven to generate overwhelming results, e.g., in XRD. The authors state "To the best of our knowledge, no spectroscopic studies have successfully analysed multi-MP mixtures". This might be true for MPs, but there are several projects applying DL-methods on spectrograms, which is the same workflow.

Secondly, the authors state "Creating sensors that can rapidly capture and detect MPs in liquid without pretreatment will accelerate ...". Regarding a possible application, I doubt the systems performance in generating high throughput results, having to purify MPs on substrates and taking spectroscopies. Wouldn't it thus be better to use simple microscopic images and apply deep learning routines like U-Nets or maskRCNNs? At least a comparison in performance would be beneficial. Furthermore, the five-class classification problem is relatively easy to solve, one should think about how to apply hundreds or thousands of chemicals.

Thirdly, and my main critics: There is a big lack of state of the arts regarding microplastic detection systems. From the article it is not clear, who else is working on the field, what the readouts and methods are, what the benchmark is, what methods use other groups etc.

Minor comments:

- Abstract does not describe the format of used data – images, spectroscopies, ...?
- Results-Section about SpecATNet contains mixture of SOTA and methods.
- Why is there a comparison of PCA vs. SpecATNet doubtful – PCA works unsupervised and therefore will always deliver worse results. The same counts for SpecATNet compared to mathematical averaging of spectra without using the self-attention function – this comparison seems to be unfair, as there are spectra not containing any information which worsen the averaging.
- I am no chemist, though the preparation of microplastic particles on diverse substrates (AgF@AgM@C10 versus AgF@AgM) seems to be arbitrary. Why exactly these substrates and no others, how do readouts behave and what needs to be improved? How robust/reproducible are readouts regarding the substrate?
- Figure S24: Leave out, does not deliver further information.
- Figure S13 and S14 let me doubt the consistency of the data – it would be beneficial to test a classifier trained on freshly prepared substrates and test it on “old” substrates. If this does not work, the system is highly sensitive to the age of the substrates.

REVIEWER COMMENTS

Manuscript ID: NCOMMS-23-32695A

Reply: Author's response to referees on paper entitled “Pay attention to microplastics: SERS sensing of microplastics using self-attention-based neural networks on hierarchically-porous Ag foams”

We deeply appreciate the reviewers and the precious time spent reading our manuscript and providing criticism. Below is our response to each of the reviewers' queries highlighted in Bold and our response in Normal font (blue color). Any changes made to the manuscript are clearly outlined in the response, and selected Figures are reproduced in this response for the convenience of the reviewers. All changes in the manuscript are highlighted in yellow.

Due to the significant number of requests for additional data, we summarized the new experiments information in the table below:

The list of additional experiments performed for the revision:

Brief description of the experiment	Added/revised figures/tables	Location of discussion in text
We modeled the permeability and fluid velocity of the macroporous foams to show how adding mesopores to the surface should lead to a greater retention of fluids and MPs and larger capillary forces.	Fig. S10	P. 9, L 182
We measured polyethylene terephthalate (PET) to our workflow and trained our NN to identify it as our 6 th type of MP and analyzed PET in different mixtures.	Fig. 5, 6 Fig. S17, S19-21, S27, S29, S34	P, 12 L 242, P. 19 L 361, P. 23, L 429
We tested low-concentration PS suspensions down to 10 P/L.	Fig. 5	P. 18, L 358
Low dispersity PE films were analyzed to demonstrate a higher variability of available MPs sizes.	Fig. 6 Fig. S29, S34	P. 21, L 393, P. 23, L 429
We used Humic Acid, Marine Sediments, Chromium-tainted soil, synthetic seawater and algae to attempt to confuse our NN and determine if it could still accurately identify MPs.	Fig. 6 Fig. S30, S32, S34	P. 21, L 396, P. 23, L423
We analyzed the MP samples on aged AgF@AgM@C10 after 1 month of storage in air.	Fig. S40	P. 26. L 501
We evaluated the cost of the AgF@AgM@C10 substrates and compared them to SERS-active porous foams from cheaper metals that we also prepared.	Fig. S41, Tab. S8	P. 27, L 519
We report the accuracy, precision, recall and F1 scores for the new experiments.	Fig. S37	P. 24, L 464
We report the precision-recall curves and receiver operating characteristic curves.	Fig. S38	P. 25, L 471

SpecATNet is compared to a logical regression model, a decision tree model and a support vector machine.

Fig. S39

P. 25, L 486

Reviewer #1 (Remarks to the Author):

R1-Q1: The main defect of this study is that they do not analyze environmental samples with their method. Additional experiments should be conducted with natural water samples and the detection effect should be presented, otherwise the application of this method will be limited to laboratory analysis of standard samples.”

R1-A1: We agree with Reviewer #1. We performed additional experiments using MPs with diverse sizes, shapes and chemical structures. We also added measurements in various water samples/matrices. More information about these experiments is described in detail below and in this table:

The list of additional experiments performed during revision

Brief description of the experiment	Added/revised figures/tables	Location of discussion in text
We modeled the permeability and fluid velocity of the macroporous foams to show how adding mesopores to the surface should lead to a greater retention of fluids and MPs and larger capillary forces.	Fig. S10	P. 9, L 182
We measured polyethylene terephthalate (PET) to our workflow and trained our NN to identify it as our 6 th type of MP and analyzed PET in different mixtures.	Fig. 5, 6 Fig. S17, S19-21, S27, S29, S34	P, 12 L 242, P. 19 L 361, P. 23, L 429
We tested low-concentration PS suspensions down to 10 P/L.	Fig. 5	P. 18, L 358
Low dispersity PE films were analyzed to demonstrate a higher variability of available MPs sizes.	Fig. 6 Fig. S29, S34	P. 21, L 393, P. 23, L 429
We used Humic Acid, Marine Sediments, Chromium-tainted soil, synthetic seawater and algae to attempt to confuse our NN and determine if it could still accurately identify MPs.	Fig. 6 Fig. S30, S32, S34	P. 21, L 396, P. 23, L423
We analyzed the MP samples on aged AgF@AgM@C10 after 1 month of storage in air.	Fig. S40	P. 26. L 501
We evaluated the cost of the AgF@AgM@C10 substrates and compared them to SERS-active porous foams from cheaper metals that we also prepared.	Fig. S41, Tab. S8	P. 27, L 519
We report the accuracy, precision, recall and F1 scores for the new experiments.	Fig. S37	P. 24, L 464
We report the precision-recall curves and receiver operating characteristic curves.	Fig. S38	P. 25, L 471
SpecATNet is compared to a logical regression model, a decision tree model and a support vector machine.	Fig. S39	P. 25, L 486

R1-Q2: The selection of five different MPs should be elaborated. We know that some other polymers such as PVC, PP and PET plastics are commonly used and detected in the environment. Why only the five polymer types are chosen?

R1-A2: Microplastic (MP) pollution is a multifaceted problem because polymer materials have different properties and sources that affect transport, environmental fate and ecological impacts. We chose these initial five polymers based on the following considerations: *(I)* they represent a diversity of industrial-derived and consumer-derived pollution sources generating primary and secondary MPs (i.e. MPs designed to be small versus fragments, respectively). *(II)* These polymers are becoming widespread in different marine/freshwater environments. MPs have different abundances but we tried to select targets with different transport properties and environmental fates. *(III)* Each polymer selected has a diversity in chemical structure, which will determine susceptibility to degradation, which might translate into overall persistence in the environment (e.g. PS>PE>Nylon>PMMA>PTFE). Also, from a practical perspective, the five polymers we tested are available in a range of *(IV)* shapes and *(V)* sizes, and many procedures exist to modify their size and surface structure. This allowed us to test the impact of chemical structure and morphology on the performance (accuracy, precision, recall, F1 score) of the neural network (NN). The detailed scheme with an explanation was added to SI – **Fig. S16** and related discussion.

Considering all of the factors above, we systematically tested the accuracy of SpecATNet in identifying 1,2,3,4, and 5 different polymer mixtures. Then we tested these samples with different polymer morphologies, interference agents, and aqueous matrices. The NN performed well at all of these multi-label classification tasks---better than any comparable work in this field---thus, we were sufficiently impressed with the accuracy of the NN and the ability of the hierarchical metal foams to trap MPs so we felt it was time to report our progress.

For example, to give the reviewer some reference of what is state-of-the-art in the MP sensing field for SERS, this paper (*Advanced Functional Materials*; 2023, 2307584; I.F. = 19) was submitted to AFM in the same month (July 2023) we submitted to *Nature Communications*. They used SERS and classified only two (2) kinds of MPs and report sensitivities down to 10 mg/L using a simple logistic regression algorithm borrowed from scikit-learn. 10 mg/L is orders of magnitude higher than typically found in marine/freshwater samples, so this technique will still likely require PCPT methods. And it can only measure small MPs (~1 to 2 μm). Meanwhile, we measure 6 MPs and our workflow can measure down to ~0.00015mg/L, which enables PCPT-free sensing. We also show that logistical regression, decision tree and support vector machine models are inferior to SpecATNet (see **Fig. S39**, P. 25, L 470).

One of the strengths of these NNs is they are upgradeable with more high-quality data. To demonstrate that SpecATNet can be upgraded with more MP types, we added a 6th polymer: polyethylene terephthalate (PET) fibers (**Figs. 5,6**).

Overall, we made the following changes:

-Added **Fig. S16** and related discussion (Page(P.) 12, Line(L.) 242) to explain our motivation in choosing these types of MPs.

-We upgraded our sensing system to enable the detection of PET fiber. The related figures were revised, namely, **Fig. 5, 6, Fig. S17, S19-21, S27, S29, S34** (P, 12 L 242, P. 19 L 361, P. 23, L 429).

R1-Q3. The size selection standard should be illustrated, and could this method detect large MPs (90-500 μm or even 500-5000 μm)?

R1-A3. Fig. S17 and Fig. S28 illustrate the size of the selected MPs. We added low dispersity PE films, PS foam fragments, and PET fibers to demonstrate higher variability of available MPs sizes. The MPs are 0.3-90 μm and PET fibers are 20 ± 2 μm in diameter with different lengths up to ~ 0.3 mm. In general, the SERS sensors used in these experiments are composed of a porous metal foam with macropores ~ 262 nm in size, which limits the maximum size of MPs that can be captured and detected.

We made the following changes:

-Added new types of samples and revised corresponding **Fig. 6, Fig. S29, S34** (P. 21, L 393, P. 23, L 429)

-We discussed size limits for the current architecture and how to potentially create new substrates using additive manufacturing techniques (P. 26. L 397, P.27 P519).

R1-Q4. We know that the shapes of MPs vary a lot, including spheres, fragments, films, fibers and foam etc. And in realistic environment, fragments and fibers account for a large proportion. However, only quasi-spherical MPs were tested in this study, some other shapes of MPs should be characterized with this method to verify the effect.

R1-A4. Reviewer #1 is correct. In addition to the degraded MPs we measured earlier, in the new experiments we also measured PET fibers and PS fragments derived from both foam and films (**Fig. S17, 28**). These results are described in **Fig. 5** and **Fig. 6**.

QR1-5. Generally, the abundance of MPs in freshwater hardly exceeds 100 items per liter. In this study, the flowed suspensions of MPs are unrealistically high (100~10000 particles per liter). How could this method be used without pre-enrichment? The enrichment method of MPs might be developed to facilitate the use of this method.

R1-A5. We agree with Reviewer #1 that the concentration of microplastics can be lower than 100 P/L, especially in more remote freshwater locations. We consider our macroporous-mesoporous Ag metal foams somewhat analogous to Neuston nets. Neuston nets are primarily used in aquatic research to passively collect organisms and particles in the uppermost layer of water bodies as the nets are towed through the water. Like Neuston nets, the collection of MPs in more pristine environments can be enhanced by increasing the flow volume and/or flow duration. We don't view increasing flow volume/duration as pre-enrichment because it is passive and relatively fast (~ 20 minutes). The AgF@AgM@C10 substrates were exposed to a 10 P/L solution of MPs for 20 minutes and were capable of sensing without significant loss in accuracy. However, a larger number of spectra is required to reach $>95\%$ accuracy, for example, <3 spectra were enough to detect 100 P/L and 20 spectra are required for 10 P/L detection. The detailed discussion is located in **Fig. 5A** and P. 18, L 356.

R1-Q6. "Although the commercial microplastics were artificially aged using oxidants and UV light, they could not mimic the environmental microplastics accurately, because there are usually organisms (microorganism, microalgae or fungi) and organic matters attached on the MPs, which might have an influence on their Raman spectrum acquisition. Moreover,

why organic material (0.3 mg/mL BSA protein) other than humic acid or fulvic acid was selected? You know that humic acid and fulvic acid are normal organic matters detected in natural water.”

R1-A6. As Reviewer #1 suggested, we could have gone a lot deeper in examining the impact of organic matter on the sensitivity and collection efficiency of the setup. To expand this section, we analyzed MPs in these additional environmental matrices:

-humic acid (0.01 mg/ml concentration)

-marine sediments (polychlorinated biphenyls and organochlorine pesticides in marine sediments NMIJ CRM-7304-a)

-soil (Chromium VI – Soil, RTC CRM041-030)

-synthetic seawater (SSWS30)

-algae (grown from Spirulina Powder)

In total, 10 additional kinds of samples were measured, and the results are presented in **Fig. 6** (*shown below*), **Fig. S29** and **Fig. S34** (P. 21, L 396, P. 23, L423). We showed that the workflow is sufficient to identify MPs in various environmental matrices.

Fig. 6. Analysis of realistic samples containing degraded MPs in different matrices. (a) Fluorescence microscopy images of AgF, AgF@AgM and AgF@AgM@C10 following protein adhesion tests using fluorescein-conjugated BSA (BSA-FITC). (b) A confusion matrix visualizing the performance of SpecATNet for the analysis of multiplexed environmental samples, wastewater, seawater and samples containing humic acid. (c) Microscope images of samples containing algae, soil and degraded MPs (c-I and c-ii) and SEM images (c-iii and c-iv) of AgF@AgM@C10 after flowing MPs in environmental matrices. (d) A confusion matrix visualizing the performance of SpecATNet for the analysis of the multiplexed samples, containing algae, soil, sediments and humic acid

R1-Q7. The deficiency or limitation of this method should be presented in the last paragraph of the results and discussion section.

R1-A7. We added a paragraph at the end of the Discussion section to explain the current limitations of the workflow and how it could be improved over time (P 26, L496).

Results and discussions.

Despite the high performance of the developed sensing procedure, there are some features that could be improved with further iterations of the SERS sensor, NN and analytical setup. They are: (1) the micropore size of the Ag foam used in this work determines the maximum size of MPs – 262 μm . However, macroporous templates with larger pore sizes or more complex networks could be generated using additive manufacturing⁵⁹. And much larger MPs are commonly determined by visual methods^{60,61}. (2) Ag does not have an indefinite shelf-life. Despite preliminary results show the accuracy in identifying MP mixtures remains consistent after one month of air storage for PTFE/PE and PMMA/PS (**Fig. S40**), further stability studies are needed. Using mesoporous Au would improve stability and enable measurements at near-infrared wavelengths. (3) Small Raman spectrometers can be made inexpensively using off-the-shelf parts (<\$3600 according to OpenRAMAN)⁶² but more work must be done to design complete optical setups that are easier to deploy even for the most resource-limited labs. (4) Upgrading of SpecATNet as with any other NNs requires the collection of a large dataset. For example, a minimum ~4000 spectra are required to introduce new MPs with different chemical structures. Collecting a large and diverse dataset of spectral data for training CNNs can be challenging, while limited data can lead to overfitting and poor generalization to new, unseen spectra.

Minor comments

Title, Pay attention to microplastics, this phrase is not highlighted the content of this study, replace or delete

We meant our title to be an homage to one of the earliest transformer-type NN papers that has the title “Attention is All you Need” (see: *Advances in Neural Information Processing Systems*; **2017**; 30). But we agree our title might not be obvious to most readers. So, we changed the title to “Pretreatment-free SERS sensing of microplastics using a self-attention-based neural network on hierarchically porous Ag foams”.

L30, real-time monitoring of MPs is hard to realize by this method, so immense potential is not appropriate.

L36, divided should be degraded

L47, and should be or

L287, delete -

L324, single MP sample

L331, varied should be changed

L334, PS-MPs

Fig.5e is not cited in the text.

L358, Fig.5b should be Fig.5e?

L359, Fig.S25b? please check this.

The errors above have been revised.

L366-367, how did you get this value 0.0015 mg/L, could you explain?

The conversion of MPs (beads) concentration was performed according to eq. on p. 26 (SI) and examples are on **Tab. S3**.

Converting concentration of MP from mg/L to Particles/L:

MP suspensions were prepared by weight concentrations (mg/L) and further recalculated to particles per liter (P/L) taking into account the size distribution (Fig. S15) according to the equation ⁹:

$$C_{P/L} = \frac{C_{mg/L} \times 10^9 (\text{unit fraction factor})}{\left(\frac{\pi}{6}\right) \times \text{density} \left(\frac{g}{cm^3}\right) \times (\sum D_n (\mu m) \times P_n (\%))^3}$$

Where D_n is the diameter of the MP bead, and P_n is the percentage content of this fraction.

L401, add microplastics after PS

L407-408, this study does not analyze really environmental samples either, please revise.

L415, from 80 to 96%

L417, delete within

Marked errors have been revised.

L431, The conclusion section is a little lengthy

The conclusion was revised.

L432-435, These sentences can be deleted

L441-445, This sentence is too long, please revise.

L453-455, it can collect numerous MPs from the environment on-site without onerous separation and analytical methods, this statement is not appropriate, because you don't really check the real effect of this method on-site without separation and analytical treatment.

The sentence was revised, on-site was deleted. We tested additional samples.

L468, Why the irradiation time was so different?

The irradiation time was chosen by a preliminary kinetics study of MP degradation for each material. PE beads were surprisingly easy to degrade, while PS, PMMA and Nylon changed morphology after ~24 hours of exposure. PTFE is an inert polymer; thus no changes were observed even after 24, 48 and 72 hours. The PTFE samples exposed for 72 h were used in the sensing experiments, although minimal morphological changes were observed.

L469, minutes should be min

L515, change x with ×

L532-L604, some contents are missing, please check

Marked errors have been revised. We are very grateful for the reviewer's comments and the thoroughness of their report.

Reviewer #2 (Remarks to the Author):

R2-Q1. For real impact, the speed, throughput and portability would be the key for sensing and classifying MPs. The presented results are very low throughput,....

R2-A1. We thank the Reviewer for the response and the opportunity to explain better our motivation and the state of high-throughput methods in MP classification. Measuring raw untreated microplastic (MP) samples from the environment is very challenging because: **(1)** MPs have similar chemical structures, which are difficult to differentiate in complex mixtures. Most plastics are carbon-based structures with properties that are determined by the arrangement of the carbon atoms, functional groups and side chains. Thus most MP sensing schemes require a pre-concentration or pre-treatment (**PC/PT**) steps to simplify the analytical measurement (Acc. Chem. Res.; 2019, 52, 4, 858–866). **(2)** MPs are much lower in concentration relative to natural organic matter, biological materials, and other anthropogenic litter (e.g. soot, oil droplets, etc). These materials also have similar chemical structures and exist in vastly higher concentrations than MPs. Before analysis, PC/PT steps are also necessary to collect enough MPs and separate them from other organic interference agents. **(3)** The combination of (1) and (2) means that **PC/PT methods are the throughput bottleneck** in existing MP sensing methods, because they typically require 12 to 24 hours before analysis can even begin.

Thus, our primary motivation is to create a novel MP sensing and classification workflow that entirely omits PC/PT methods, which should dramatically improve throughput.

To illustrate this point we examined the collection, measurement and classification workflow of pyrolysis gas chromatography mass spectroscopy (pyr-GCMS) which is the gold standard in MP detection (**Fig. S36**). Creating a sample that can be loaded in the pyr-GCMS requires pre-concentration and pre-treatment steps that take up to 12 hours to extract the MP material from the environmental matrix. Mass spectroscopy requires similar timeframes (20 to 80 minutes), while interpretation and classification largely depend on the complexity of the sample and the skill of the operator. Thus we estimate that **pyr-GCMS can achieve roughly ~0.1 predictions per hour** (omitting essential steps such as interpretation and classification). By comparison, the average time required to collect samples in our workflow is 10 to 20 minutes. The sample is measured in Raman and SpecATNet takes <1 minute to process the spectra and classify the sample. Thus the throughput of one of **our measurements is about 2 to 4 predictions per hour or roughly 20 to 40 times faster than pyr-GCMS**, mainly because we do not need to perform onerous and time-demanding PCPT procedures.

R2-Q2.are acquired by rather expensive bench to P/Large equipment,

R2-A2. Regarding equipment and portability, we mainly view this as an engineering problem because small Raman spectrometers can be built using open-source plans for under \$4,000 that generate high-resolution scans at short acquisition times and can be adapted to match the key features of our setup. For a rough estimate of the cost and form factor for a DIY small Raman setup, see the OpenRAMAN project (<https://www.open-raman.org/build/>). Obviously, we need to do some work on this point, but we think this part of the effort is a straightforward engineering problem and hope that the publication of our manuscript will inspire ingenious builders in the field to create smaller and cheaper Raman spectrometers using off-the-shelf parts.

R2-Q3.and it is practically impossible for this technique in its current form to screen large volumes of samples rapidly in a practical setting.

R2-A3. In light of the above information, it should be reasonable to expect that Raman is much more plausible as an effective, portable and inexpensive workflow for MP detection, especially since we demonstrated that it can omit PCPT methods with some assistance from self-attention-based NNs. Other methods like pyr-GCMS use bulky instruments that are difficult to miniaturize with off-the-shelf parts, and visual identification techniques are less accurate as the MPs decrease in size. For a comparison of the various MP sensing techniques see **Fig. S36** and the discussion (P. 24 L. 454)

As we mentioned above, our workflow has roughly 20- to 40-times higher throughput than equivalent MP sensing and classification methods. Our SERS substrates are very inexpensive, and in a separate note we describe how the cost of the sensor can be driven \$0.5 per unit. Please see **Fig. S41**, **Tab. S8** and related discussion in SI.

In our opinion, low-cost sensors are the key to screening large volumes of MPs because each substrate is capable of omitting PCPT protocols unlike other methods. Low cost means that these samples are easy to deploy in the field and then collect for sensing and classification which is rapid using open-source methods like SpecATNet (<https://github.com/Tre1725/SpecATNet>). As the Raman spectrometers become more inexpensive and miniaturized, it is reasonable to expect these methods to enable in-field MP classification shortly. And this is possible because they omit PCPT methods from the workflow.

R2-Q4. Although valuable in their discipline, the results belong to a more specialized journal.

A4. To give the reviewer some reference of what is state-of-the-art in the MP sensing field for SERS and the magnitude of our accomplishments, this paper (*Advanced Functional Materials*; 2023, 2307584; I.F. = 19) was submitted to AFM in the same month (July 2023) we submitted to *Nature Communications*. They used SERS and classified only two (2) kinds of MPs and report sensitivities down to 10 mg/L using a simple logistic regression algorithm borrowed from scikit-learn. 10 mg/L is orders of magnitude higher than typically found in marine/freshwater samples, so this technique will still likely require PCPT methods. And it can only measure small MPs (~1 to 2 μm). Meanwhile, we measure 6 MPs and our workflow can measure down to ~0.00015mg/L, which enables PCPT-free sensing. We also show that logistical regression, decision tree and support vector machine models are inferior to SpecATNet (see **Fig. S39**, P. 25, L 470).

As we explained above, throughput is the main challenge in all MP sensing methods. And throughput is primarily determined by PCPT methods. We attempted to explain the challenges in this field better and describe our motivation for PCPT-free sensing and its impact on the broader effort to detect MPs from various MP pollution sources (**Fig. S16** and related discussion). We also added **Tab. S1** describing the different methods used to identify MPs and compared them in terms of ease of use and cost and revised introduction accordingly (**Tab. S1**, P. 3 L. 60).

We achieved PCPT-free sensing of MPs by creating a material that has sufficient affinity for MPs to collect them from flowing solutions while demonstrating built-in sensing ability via SERS. But this was not enough to omit PCPT methods. We also had to develop a type of NN that borrows a self-attention-based mechanism from new natural language processing algorithms to accurately analyze the data and achieve exceptional performance over 6 kinds of MPs with various sizes and

shapes in a range of interference agents. We cannot find any existing work in the MP literature that is on the same level, so we still think sending our manuscript to this journal was appropriate.

To sum up **Questions 1-4**, we did the following changes:

-created **Tab. S1** (shown below) overviewing the most important parameter of MPs detection strategies (P. 3 L. 60)

-calculated throughput of SpecATNet+SERS and compared it to Pyr-GCMS (**Fig. S36**, discussion on p. 24, L. 454, shown below)

-calculated the cost to prepare AgF@AgM@C10 and showed potential to decrease it by preparing 3D porous SERS substrates from cheaper metals (**Fig. S41**, Tab. S8, p. 27. L. 519)

a Pyr-GCMS

b SERS- SpecATNet

Figure S36. Timeline of a) Pyr-GCMS and b) SERS-SpecATNet -assisted analysis of environmental MPs samples

Pyr-GCMS is a powerful technique, but it requires pretreatment by filtration, digestion, or chemical extraction because the environmental matrix produces interfering signals (**Tab. S1**). The interpretation of mass spectra of MP is already challenging and requires chemical derivatization, and interfering signals make analysis more difficult. From the literature survey^{9,20}, the pretreatment takes more than 12 hours, which is the bottleneck in sensor throughput. The Pyr-GCMS data analysis has not been automatized by ML probably due to differences in mass spectra depending on sample type. Therefore, it was impossible to evaluate this step. To sum up, sensor throughput was tentatively evaluated as 0.1 prediction/hour.

SERS-SpecATNet needs 20 minutes to enrich the sensor by the flowing water sample through porous AgF@AgM@C10, this step can be performed simultaneously on multiple sample to accelerate the process.

Sensor enrichment is followed by collection of SERS dataset from sample, where 20 spectra are minimum required taking 3 minutes (10 minutes for 60 spectra). The processing of dataset takes less than a minute, giving 2-4 predictions/ hour, which is at least $\times 20$ times faster than Pyr -GCMS and more suitable for in a practical setting.

Table S1. Comparative table of visual, mass and optical spectroscopic strategies for MPs analysis

	General	Representative example	General	Representative example	General	Representative example
	Visual		Mass		Optical spectroscopy	
Examples	Naked eye, digital camera, microscopy (light, dissecting, confocal, scanning electron, atomic force)		Pyr-GCMS, TGA-MS/DSC		FTIR, Raman, SERS	
Chemical analysis	no	-	yes	PE, PP,PS, PET, PVC, PC, PMMA ¹⁶ ; PET ¹⁷	yes	CA, PS, PE, PP, PA ¹⁸ ; PVC, PU, PS, PP, PE, PMMA ^{19,20}
Detection of MP mixture	No (only visual)	^{21,22}	yes	¹⁶	no (1 example)	²³
Size range	yes	Digital camera – 1 mm ²¹ ; Fluorescent ²⁴ – 100 μm	No	single particles measurement 50 μm ²⁵ ; 0.1 μm ¹⁶	yes	FTIR – >10-20 μm Raman- >1 μm SERS >360 nm ²⁶
Environmental sample analysis	Yes	Marine sample ²¹ ; Marine sediments ²⁷	Yes	cellulose, pine wood, wool and cotton ²⁸ ; sediment sample ²⁹	Yes	Marine sample ^{19,23} Wastewater ³⁰
Pretreatment	Required	density separation, staining ²¹ ; sieving, manual inspection ³¹ ; chemical digestion ^{27,32} ; extraction by organic solvent ³³	Required	chemical and enzymatic digestion, derivatization ^{28,29}	Required	Filtering, centrifugation, H ₂ O ₂ , NaI ³⁴ Manual selection ²³
Portability	Yes (large MP)	Digital camera – 1 mm ²¹	No	³⁵	Yes	smartphone-based Raman analyzer ³⁶ ; portable Raman spectrometer ³⁷
Operator qualification	Low	²¹	High	^{16,25}	Moderate-Low	^{36, 37}
Price of analysis per sample	Cheap-moderate	Camera, common microscopes – cheap; Other microscopies - moderate	High	$\approx 1000 \text{ €}$ ³⁸	Cheap-moderate	$\approx 300 \text{ €}$ ³⁹

Price of equipment		From \approx 1000 €		From \approx 100 000 €		From \approx 3700 €
Combination with NNs	Yes	21,22	No	-	Yes	18-20,40

R2-Q5. Several clarifications are needed regarding the distribution of the training and test datasets. The authors stated that the datasets were divided in a 10-fold cross-validation manner, and detailed numbers of various sample types are provided in Supplementary Table S6. However, certain sample types, such as PS(AgF@AgM@C10), Nylon(AgF@AgM@C10), and PMMA+PS, have only one sample. This raises the question of how SpecATNet could accurately predict these types when not included in the training set. Could it be that the spectral distributions of these samples were embedded/hidden within the spectra of other MP combinations, allowing for accurate identification even without being included in the training? Additional analysis and explanations on this matter are needed. Furthermore, Table S6 listed some sample types twice, such as PTFE(AgF@AgM@C10) and PMMA+PTFE, but with different sample counts. The authors should elucidate why these sample types are duplicated and specify the distinctions between each instance.

R2-A5. We appreciate the reviewer spotting the duplicates. Due to the requested additional experiments, **Tab. S6** was revised with all duplicates eliminated. Also, all collected spectra are available at [<https://www.kaggle.com/datasets/andriitrelin/microplastics-raman-spectra>].

To reply to the second part about the possibility of multi-label analysis of samples potentially not being included in the training set, we would point their attention to i) our relatively balanced training set, and description of ii) our strategy towards training/identification of various MPs combinations. First, we attempted to obtain a similar number of samples (and corresponding spectra) with each polymer independently. In total, for PE – 12018 spectra, PMMA – 14105 spectra, PTFE – 7179 spectra, PS – 13116 spectra, Nylon – 8591 spectra, PET -3632 spectra. Therefore, we avoided the case when there is a vanishingly small number of any of the polymers, at least 12% of spectra for each polymer, present in the dataset. In general, dataset can be considered as balanced because we tested impact of oversampling as a balancing technique on model performance and found that balancing did not effect model performance. Therefore, balancing was omitted in the subsequent experiments.

Secondly, we believe that SpecATNet can accurately predict polymer mixtures that have not been observed during training. One can think of multilabel classification as a set of independent binary classification tasks. Each classifier determines the presence or absence of a single polymer. From this point of view, training the model on pure polymers alone should be enough to identify all possible mixtures. However, in practice, the model also needs training to disentangle mixed spectra, meaning that mixtures must be part of the training dataset. Once trained on pure polymers and mixtures, the model can successfully predict previously unobserved polymer mixtures, extending its knowledge from pure polymers and some mixtures to other mixtures.

R2-Q6. In the performance analysis of SpecATNet, the authors could include a negative control scenario. Specifically, they could test samples that contain no MPs but include other materials and some potential particles commonly found in realistic environmental water samples—to assess whether the detection of system generates false positives. This additional analysis would offer a more complete understanding of the system’s accuracy.

R2-A6. We followed Reviewer’s recommendations and included 8 sample types without MP: (1) sensor surface (2) wastewater, (3) algae suspension, (4) protein solution, (5) humic acid solution, (6) seawater, (7) soil (8) and sediments. For all samples, SpecATNet made correct predictions about the absence of MPs with >99.5% accuracy, showing a very low false-positive response. The results are presented in **Fig. S34b** (P. 23 L. 433). Moreover, the overall performance of the systems was re-evaluated considering all new measurements and presented on **Fig. S37**.

Figure S34. (a) Accuracies of SpecATNet prediction for multicomponent samples in complex matrices **(b)** matrices without MPs

$$\text{Precision} = \frac{\text{True}_{\text{positive}}}{\text{True}_{\text{positive}} + \text{False}_{\text{positive}}} \quad
 \text{Recall} = \frac{\text{True}_{\text{positive}}}{\text{True}_{\text{positive}} + \text{False}_{\text{negative}}} \quad
 \text{F1 score} = \frac{2 \times (\text{Recall} \times \text{Precision})}{\text{Recall} + \text{Precision}}$$

Figure S37. (a) Overview of the precision, recall and F1 score based on all samples analyzed in this study on AgF@AgM@C10, **(b)** schematic explanation of true positive true negative false positive and false negative

R2-Q7. To interpret the predicted scores (decimals from 0 to 1) generated by SpecATNet, a specific threshold is required to determine whether a test sample contains a particular type of MP or not. However, the main text currently lacks this information. The authors should clarify the threshold value used and provide the rationale for its selection. ROC curves are completely missing.

R2-A7. We added the ROC curve to SI (**Fig. S38b**) and related discussion to the main text (P. 25, L 468). In addition to ROC, we show precision-recall curves (**Fig. S38a**, P. 25 L. 472) to complement the information about the tradeoff between precision and recall for different thresholds. For overall predictions, a noninformative threshold of 0.5 was used in calculating all metrics, requiring binarization of the score since no assumption about the price of type I/type II errors was made.

However, the threshold can be customized: dynamic or adaptive thresholds can be applied based on the characteristics of each microplastic type. For example, the following factors can be considered: i) toxicity of polymer type to animals/humans and ecology, ii) local knowledge about the prevalence and distribution of specific microplastic types in a particular environment, iii) research goal – if the primary goal is to study the abundance of a specific microplastic type, a lower threshold may be used to ensure comprehensive detection, iv) regulatory requirements in a specific location.

The discussion about ROC and PR curves was added on p. 25, L468.

Figure S38. (a) Precision-recall curve and (b) Receiver operating characteristic curve for six types of MPs

Minor comments:

(A) A formatting issue exists between lines 531 and 663 in the “SpecATNet architecture” subsection of the Methods section, resulting in the missing of several sentences.

(B) In line 359, “Fig. 25b” should be corrected to “Fig. S25b”.

(C) The definition of “precision” in Fig. S32 is not correct now.

(D) Some supplementary tables/figures were not cited in the main text, such as Fig. S2, Fig. S11.

(E) The color bars in Fig. 5d, Fig. 5e and Fig. 6b are not matched with the visualizations of their confusion matrixes, where green color means 1 in the confusion matrix but corresponds to low probability in the color bar.

We thank the Reviewer for all the constructive comments and the attention to detail. The marked errors have been revised.

Reviewer #3 (Remarks to the Author):

R3-Q1. Firstly, I am not sure what the novelty of the approach truly is. Material-preparation is state of the art, spectroscopic read-outs for this kind of problem is state of the art, neural networks are just used and not newly developed. Further, there is a big lack of motivation,

R3-A1. We thank the reviewer for the response and opportunity to better explain our motivation. Measuring raw untreated microplastic (MP) samples from the environment is very challenging because: (1) MPs have similar chemical structures, which are difficult to differentiate in complex mixtures. Most plastics are carbon-based structures with properties that are determined by the arrangement of the carbon atoms, functional groups and side chains. Thus most MP sensing schemes require a pre-concentration or pre-treatment (PC/PT) steps to simplify the analytical measurement (*Acc. Chem. Res.*; 2019, 52, 4, 858–866). (2) MPs are much lower in concentration relative to natural organic matter, biological materials, and other anthropogenic litter (e.g. soot, oil droplets, etc). These materials also have similar chemical structures and exist in vastly higher concentrations than MPs. PC/PT steps are also necessary to collect enough MPs and separate them from other organic interference agents prior to analysis. (3) The combination of (1) and (2) means that PC/PT methods require a lot of time and the bottleneck in existing MP sensing methods. This is in part because chemical digestion must be done relatively gently over 12 to 24 hours to ensure that the chemical structure is not modified.

Thus, our first motivation was to create a novel MP sensing and classification workflow that omits PC/PT methods entirely. The second motivation was to create an MP sensing and classification platform that is sufficiently low-cost so that it can be available even in resource-limited labs.

R3- Q2. ...it is not described why exactly the used substrates were applied,

R3-A2. To realize our goal of omitting PC/PT methods, we needed to create a material with a sufficient affinity for MPs while demonstrating some built-in sensing ability. MPs are organic macromolecules that tend to be moderately hydrophobic and are sufficiently large enough to be governed by capillary forces. Researchers in aquatic research fields frequently use a physical apparatus called a “Neuston Net” to passively collect organisms and particles in the uppermost layer of water bodies via a combination of chemical affinity and capillary forces. But Neuston nets have no sensing ability so these particles are subjected to laborious PC/PT methods and then passed to some analytical method like visual detection or pyrolysis-gas chromatography mass chromatography (pyr-GCMS).

Light can excite collective excitations of electrons on the surface of metals called surface plasmons (SPs). The electric fields associated with SPs can amplify the Raman cross-sections of adjacent molecules in a process called surface-enhanced Raman spectroscopy (SERS) that is extremely sensitive—even down to the single-molecule level (*JACS*; 2009, 131, 14466-14472). But the SP is a surface-trapped evanescent wave, so it does not extend far away (<100nm) from the metal surface thus they are not very effective at exciting large macromolecules. Planar grating structures used in SERS are also not very effective at capturing large macromolecules like MPs.

We used the Ag foam substrates because the interplay of macropores, mesopores and the hydrophobic creates a tortuous network that can trap MPs from aqueous solutions via chemical affinity and capillary forces. The 3D metal surface surrounds the MPs and enables the SP and scattered light to excite from all sides, generating a stronger SERS signal. Thus we succeeded in creating a substrate that had sufficient chemical affinity for MPs and built-in sensing ability.

But to totally omit the PC/PT steps we had to make sense of this SERS data. We devised an attention-based NN called SpecATNet to label the spectra and make definitive judgments on the contents of the samples.

R3-Q3. ...why the readout of the system is spectroscopic...

R3-A3. We didn't see a viable route to PC/PT-free methods using other types of readouts (e.g. visual, mass, *etc*). In addition, the accuracy of purely visual methods face several challenges because of resolution limits and the visual similarity of other MPs and organic/inorganic matter as the particles become smaller. Thus visual methods become increasingly subjective and inconsistent as the particle size decreases (*Chemosphere*, 2022, 308, 136449). And mass-based methods like pyr-GCMS require expensive equipment (>\$100K) that is unlikely to become portable in the near future. We included a comparison of visual, mass and optical spectroscopy approaches in **Tab. S1**.

We identified SERS as the best candidate for a PC/PT-free protocol because: (1) It is sufficiently sensitive (*JACS*; 2009, 131, 14466-14472). (2) The sensors can be fabricated cheaply. We included a separate note showing how we modified the sensor to drive the cost per sensor <\$0.5. Please see **Fig. S41, Tab. S8** (related discussion in SI) and the discussion on P. 27, 519 (3) Spectroscopic data can be rapidly analyzed with machine learning and take advantage of new self-attention-based NNs. (4) The equipment can be inexpensive and portable. We mainly see this as an engineering problem because Raman spectrometers can be built using open-source plans for under \$4,000 that generate high-resolution scans at short acquisition times and can be adapted to match the key features of our setup. For a rough cost estimate of a DIY small Raman setup, see the OpenRAMAN project (<https://www.open-raman.org/build/>). Obviously, we need to do some work on point (3), but we think this part of the effort is straightforward and hope our manuscript will inspire the ingenious builders in the field to create smaller and cheaper Raman spectrometers using off-the-shelf parts. Ultimately, we think making a low-cost sensor for resource-limited settings will have more impact on MP detection than high-throughput methods done in centralized labs.

R3-Q4. ...and why not other neural nets are used.

R3-A4. Potentially, other types of NNs could be used. But common CNNs cannot achieve high accuracy in our samples because of the nature of our data – the SERS spectra are highly inhomogeneous over the analyzed area (**Fig. S9**). The averaging of all spectra collected from SERS sensor would include spectra of Ag surface itself without MP and/or some other absorbed organic matter contained the matrix. Thus, considering the total spectra without curation would generate a less accurate classification. Therefore, the self-attention mechanism was integrated into CNN architecture to provide curation and highlight only MP spectra by assigning them a high attention weight in the classification layer. The importance of self-attention for complex MPs was visualized on **Figs. S27, 35b** and upgrade the accuracy to ~10%. Moreover, SpecATNet performs better than the logistic regression, decision tree and support vector machine models (**Fig. S39**, P. 25, L 486).

The ultimate goal of our work is to omit PC/PT methods in MP sensing. If other NN architectures could achieve this goal, they would also be suitable. However, we think self-attention is a key component in NNs going forward in AI-aided chemical analysis, especially for methods like SERS.

R3-Q5. Especially in material sciences, simple CNNs have proven to generate overwhelming results, e.g., in XRD. The authors state “To the best of our knowledge, no spectroscopic studies have successfully analysed multi-MP mixtures”. This might be true for MPs, but there are several projects applying DL-methods on spectrograms, which is the same workflow.

R3-A5. The point of our work is to create a MP sensor that does not require PC/PT steps. We tested a simple CNN but it was insufficient. Adding a self-attention mechanism markedly improved MP classification to ~10% (Figs. S27, 35b). Self-attention has become a central component in deep learning (DL) in the last ~5 years because it grants models more contextual understanding of the input data. SERS spectra in particular is high context because both the intensity and position of the vibrational peaks depend sensitively on numerous factors including the orientation of the molecule relative to the local energy flux of the electric field and proximity to the metal surface. Other projects using simple CNNs might also benefit from self-attention, especially if their data is highly contextual.

R3-Q6. Secondly, the authors state “Creating sensors that can rapidly capture and detect MPs in liquid without pre-treatment will accelerate ...”. Regarding a possible application, I doubt the systems performance in generating high throughput results, having to purify MPs on substrates and taking spectroscopies. Wouldn't it thus be better to use simple microscopic images and apply deep learning routines like U-Nets or mask RCNNs? At least a comparison in performance would be beneficial.

R3-A6. We estimated the throughput of our SERS-SpecATNet system versus pyr-GCMS (See Fig. S36; Tab. S1, P. 24, L 454). By simply eliminating the PC/PT step from our workflow we can generate 2 to 4 predictions per hour, which is roughly ×20 faster than a model workflow using pyr-GCMS. Since all other workflows require some form of PC/PT we concluded that existing methods/workflows—including visual microscopy—could not compete with plasmon-enhanced spectroscopic methods in terms of throughput.

The challenge with SERS is that the data is highly contextual in complex samples, which makes it difficult to identify MPs. Microscope images does not provide information about chemical composition of microplastics, so are unsuitable for the given task (detect presence and kind microplastic). However, there are probably other DL routines that might improve prediction even more. But we observed a great improvement in performance by adding self-attention such that it enabled the identification of MPs without any PC/PT steps, which is the main innovation of our work. But we agree with Reviewer#3 that a comparison of performance with other prediction methods would be beneficial. Therefore we added a comparison of the F1 scores of SpecATNet with linear regression, decision tree, support vector machine models (Fig. S39 and p. 25, L. 486). SpecATNet shows superior performance compared to other methods.

R3-Q7. Furthermore, the five-class classification problem is relatively easy to solve, one should think about how to apply hundreds or thousands of chemicals.

R3-A7. To give reviewer some reference of what is the state-of-the-art in the MP sensing field for SERS, this paper (*Advanced Functional Materials*; 2023, 2307584; I.F. = 19) was submitted to AFM in the same month (July 2023) we submitted to *Nature Communications*. They used SERS and classified only two (2) kinds of MPs and report sensitivities down to 10 mg/L using a simple logistic regression algorithm borrowed from scikit-learn. 10 mg/L is orders of magnitude higher

than typically found in marine/freshwater samples, so this technique will still likely require PCPT methods. And it can only likely measure small MPs (~1 to 2 μm). Meanwhile, we measure 6 MPs and our workflow can measure down to ~0.00015mg/L, which enables PCPT-free sensing. We also show that logistical regression, decision tree and support vector machine models are inferior to SpecATNet (see **Fig. S39**, P. 25, L 470).

Thus making a sensor that could detect hundreds or thousands of chemicals including MPs without PC/PT steps would be the holy grail in sensing and far beyond even the most frontier state-of-the-art in MP sensing. To demonstrate that SpecATNet is upgradable, we added polyethylene terephthalate (PET) as a 6th polymer.

To sum up **Questions 6 and 7**, we did the following changes:

-created **Tab. S1** overviewing the most important parameter of MPs detection strategies (P. 3 L. 60)

-calculated throughput of SpecATNet+SERS and compared it to Pyr-GCMS (**Fig. S36; Tab. S1**, P. 24, L 439)

-we compared the performance of SpecATNet with linear regression, decision tree, support vector machines (**Fig. S39** and P. 25, L 454)

-we upgraded our sensing system with possibility to detect PET fiber. The related figures were revised, namely, **Fig. 5, 6, Fig. S17, S19-21, S27, S29, S34** (P, 12 L 242, P. 19 L 361, P. 23, L 429)

a Pyr-GCMS

b SERS- SpecATNet

Figure S36. Timeline of a) Pyr-GCMS and b) SERS-SpecATNet -assisted analysis of environmental MPs samples

Pyr-GCMS is a powerful technique, but it requires pretreatment by filtration, digestion, or chemical extraction because the environmental matrix produces interfering signals (**Tab. S1**). The interpretation of mass spectra of MP is already challenging and requires chemical derivatization, and interfering signals make analysis more difficult. From the literature survey ^{9,20}, the pretreatment takes more than 12 hours, which is the bottleneck in sensor throughput. The Pyr-GCMS data analysis has not been automatized by ML probably due to differences in mass spectra depending on sample type. Therefore, it was impossible to evaluate this step. To sum up, sensor throughput was tentatively evaluated as 0.1 prediction/hour.

SERS-SpecATNet needs 20 minutes to enrich the sensor by the flowing water sample through porous AgF@AgM@C10, this step can be performed simultaneously on multiple sample to accelerate the process. Sensor enrichment is followed by collection of SERS dataset from sample, where 20 spectra are minimum required taking 3 minutes (10 minutes for 60 spectra). The processing of dataset takes less than a minute, giving 2-4 predictions/ hour, which is at least $\times 20$ times faster than Pyr -GCMS and more suitable for in a practical setting.

Figure S39. Comparison of F1 scores obtained from logical regression model (Log Reg), decision tree model (Dec Tree), support vector machine (SVM) and SpecATNet. The romb represent average F1 score for 6 MPs types

R3-Q8. Thirdly, and my main critics: There is a big lack of state of the arts regarding microplastic detection systems. From the article it is not clear, who else is working on the field, what the readouts and methods are, what the benchmark is, what methods use other groups etc.

R3-Q8. There are limits on total words in the manuscript, so we perhaps focused too much on conveying the main point of eliminating PC/PT methods in the earlier draft. But we can understand the Reviewer's point of view that a more detailed explanation of the state of the art is needed. Thus we expanded the introduction (P. 3, L. 60) to explain the current state-of-the-art in MP sensing and in particular we created **Tab. S1** to describe existing methods. Furthermore, in Q1 we described in detail the motivation for readout, substrate and architecture of NN.

Minor comments:

Comment 1 (C1). Abstract does not describe the format of used data – images, spectroscopies, ...?

Answer to Comment 1 (AC1). We revised the abstract.

“We developed a neural network (NN) algorithm called SpecATNet that employs a self-attention mechanism to enable the NN to learn the complex dependencies and patterns in SERS data to identify six common types of MPs (polystyrene, polyethylene, polymethylmethacrylate, polytetrafluoroethylene, nylon, polyethylene terephthalate) inside porous Ag substrates using optical spectroscopy readout.”

C2. Results-Section about SpecATNet contains mixture of SOTA and methods.

AC2. The introduction explains current MP sensing methods and the motivation for PCPT-free MP sensing using a combination of SERS with tortuous metal networks and the use of self-attention-based NNs to interpret the data. We explained the finer details of the NN in the discussion and methods in order to make it clear to the reader the contextual nature of SERS data and how SpecATNet analyzes it. We believe that SpecATNet or other NN with self-attention can be transferred and applied to other spectroscopic data, therefore providing explanations of logic flow in the related section.

C3. Why is there a comparison of PCA vs. SpecATNet doubtful – PCA works unsupervised and therefore will always deliver worse results. The same counts for SpecATNet compared to mathematical averaging of spectra without using the self-attention function – this comparison seems to be unfair, as there are spectra not containing any information which worsen the averaging.

AC3. We revised our strategy toward the comparison of SpecATNet with PCA. Instead of PCA, we compared with linear regression, decision tree, support vector machines (**Fig. S39**, P. 25 L. 486). One might think about creating alternative approaches, more efficient than averaging (e.g. comparison to the templates followed by thresholding), but our goal was to demonstrate that we

can achieve improvement of prediction quality in a purely data-driven manner, without adding any explicit filtering algorithms.

Regarding the comparison with NN using the self-attention function –we disagree because in collected spectra there could be many spectra of AgF@AgM@C10 and components of environmental matrices, such as BSA, inorganic salts (seawater, soil with high concentration of soil), algae, humic acid, soil and sediments component. SpecATNet is trained to give low weight to these spectra to improve the accuracy of M predictions.

C4. I am no chemist, though the preparation of microplastic particles on diverse substrates (AgF@AgM@C10 versus AgF@AgM) seems to be arbitrary. Why exactly these substrates and no others, how do readouts behave and what needs to be improved? How robust/reproducible are readouts regarding the substrate?

AC4. We thank the reviewer for these comments because it is critical that we craft text that is clear and understandable to non-chemists. We explained why we designed these substrates in part #2 above: (“...it is not described why exactly the used substrates were applied”).

The motivation was summarized in the introduction part. Later, the comparison of our workflow with other SERS substrates was shown in **Tab. S5**, and there is a related discussion on P 23, L. 443.

C5. Figure S24: Leave out, does not deliver further information.

AC5. Figures in SI were reorganized.

C6. Figure S13 and S14 let me doubt the consistency of the data – it would be beneficial to test a classifier trained on freshly prepared substrates and test it on “old” substrates. If this does not work, the system is highly sensitive to the age of the substrates.

AC6. To test the applicability of stored substrates, as prepared AgF@AgM@C10 was stored on air at ambient conditions. After 1 month, PTFE/PE and PMMS/PS were analyzed again using the same procedure. No considerable difference has been identified from **Fig. S40**. The related discussion was added to the manuscript on P. 25, L. 501.

Results and discussion.

The second limitation is the storage-dependent stability of Ag. Despite preliminary results showing that the accuracy in identifying MP mixtures remains consistent after one month of air storage for PTFE/PE and PMMA/PS (**Fig. S40**), further stability studies are needed.

Figure S40. Comparison of freshly prepared and after 1 month air storage AgF@AgM@C10 performance for sensing MPs mixtures

REVIEWERS' COMMENTS

Reviewer #1 (Remarks to the Author):

The authors have addressed all the comments, and they have conducted additional experiments, now I suggest it could be accepted for publication.

Reviewer #2 (Remarks to the Author):

The authors' thorough revisions have addressed most of the referee comments and have significantly improved their manuscript.

Reviewer #3 (Remarks to the Author):

The authors answered my questions.